# An integrated multi-omics approach reveals polymethoxylated flavonoid biosynthesis in *Citrus reticulata* cv. Chachiensis

Jiawen Wen [1,2,6], Yayu Wang [1,6], Xu Lu [3,6], Huimin Pan [4,6], Dian Jin[3], Jialing Wen[1,2], Canzhi Jin[1,2], Sunil Kumar Sahu [1,2], Jianmu Su [4], Xinyue Luo[1,2], Xiaohuan Jin[1], Jiao Zhao[1], Hong Wu [4] ✉, E-Hu Liu [5] ✉ & Huan Liu [1] ✉

*Citrus reticulata* cv. Chachiensis (CRC) is an important medicinal plant, its dried mature peels named "Guangchenpi", has been used as a traditional Chinese medicine to treat cough, indigestion, and lung diseases for several hundred years. However, the biosynthesis of the crucial natural products polymethoxylated flavonoids (PMFs) in CRC remains unclear. Here, we report a chromosome-scale genome assembly of CRC with the size of 314.96 Mb and a contig N50 of 16.22 Mb. Using multi-omics resources, we discover a putative caffeic acid *O*-methyltransferase (CcOMT1) that can transfer a methyl group to the 3-hydroxyl of natsudaidain to form 3,5,6,7,8,3',4'-heptamethoxyflavone (HPMF). Based on transient overexpression and virus-induced gene silencing experiments, we propose that CcOMT1 is a candidate enzyme in HPMF biosynthesis. In addition, a potential gene regulatory network associated with PMF biosynthesis is identified. This study provides insights into PMF biosynthesis and may assist future research on mining genes for the biosynthesis of plant-based medicines.

In China, the dried mandarin peel called "Chenpi" (Citri Reticulatae Pericarpium), has been used for disease treatment for two thousand years, and dates back to the Han dynasty, according to the earliest work on Chinese medicine *Shennong Bencao Jing*. *Citrus reticulata* cv. Chachiensis (CRC), also known as 'Chachi', is one of the early cultivated varieties of *C. reticulata*, which is mainly grown in Xinhui, Guangdong province, China. The dry and ripe peel of CRC named "Guangchenpi" is considered as the best "Chenpi"[1,2]. The potential medicinal uses of "Guangchenpi" includes treating cough, digestive syndrome, and lung diseases[3]. Flavonoids are one of the main medicinal ingredients of "Guangchenpi", and polymethoxylated flavonoids (PMFs) with four or more methoxy groups are the key component that can distinguish

"Guangchenpi" significantly from "Chenpi"[4]. PMFs possess a wide range of bioactivities, including potential anticancer[5,6], neuroprotective[7], anti-inflammatory[8], anti-obesity[9,10], antioxidant[11], antiatherosclerotic[12,13] effects. In particular, 3,5,6,7,8,3',4'-heptamethoxyflavone (HPMF) has been reported to exhibit antitumor initiating activity, and could be a potential candidate medicine for the treatment of cancer[14–16]. However, despite the increasing interest in "Guangchenpi", there is still a limited understanding of the biosynthesis of PMFs in citrus fruits.

Flavonoids are enriched in oranges and mandarins, which contain a large group of compounds and can be divided into several different subgroups according to their structures, including chalcones, flavones, flavanones, flavonols, isoflavonoids, anthocyanins, condensed tannins

[1]State Key Laboratory of Agricultural Genomics, Key Laboratory of Genomics, Ministry of Agriculture, BGI Research, Shenzhen 518083, China. [2]College of Life Sciences, University of Chinese Academy of Sciences, 100049 Beijing, China. [3]State Key Laboratory of Natural Medicines, School of Traditional Chinese Pharmacy, China Pharmaceutical University, No. 24 Tongjia Lane, Nanjing 210009, China. [4]Guangdong Laboratory for Lingnan Modern Agriculture, College of Life Sciences, South China Agricultural University, Guangzhou 510642, China. [5]School of Pharmacy, Nanjing University of Chinese Medicine, Nanjing 210023, China. [6]These authors contributed equally: Jiawen Wen, Yayu Wang, Xu Lu, Huimin Pan. ✉e-mail: wh@scau.edu.cn; liuehu2011@163.com; liuhuan@genomics.cn

and aurones[17]. Without flavonoids, plants cannot grow or adapt to their surroundings. Plants manufacture flavonoids to guard against biotic stress, such as insect attack and pathogen infection[18,19], as well as abiotic stress, such as UV radiation[20,21]. PMFs are special kinds of flavonoids, with more than three methoxy groups and primarily exist in the form of glycosides. In the PMF biosynthetic process, naringenin is the dominant group of flavonoid precursors mainly regulated by two key enzymes, chalcone synthase (CHS) and chalcone isomerase (CHI). PMFs are derived from naringenin with several steps of modifications. There was a complicated correlation between the number/position of methoxy groups and anticancer activity of PMFs. With the increase of methoxy groups, PMFs exhibit greater hydrophobicity when approaching and penetrating cancer cells, and thus have greater biological activity[22]. *O*-methyltransferase (OMT) is a key enzyme in the PMF biosynthetic pathway that transforms the hydroxy groups of flavonoids into methoxy groups. Generally, there are two large groups of OMTs in plants. One group is caffeic acid *O*-methyltransferase (COMT), which has a wide range of substrates, including myoinositol, chalcones, and caffeic acid. The other group is caffeoyl-CoA *O*-methyltransferase (CCoAOMT), which needs ionic cofactors in the reaction[23,24]. Many studies have tried to elucidate the key OMTs involved in PMF biosynthesis. However, very few OMTs have been identified. In the citrus variety Ougan, the CCoAOMT-like enzyme CrOMT1 can catalyze flavones containing 6-OH- and 8-OH- with adjacent hydroxy moieties[25]. Similarly, CitOMT can catalyze 3′-OH into 3′-OCH$_3$ when neighboring hydroxy moieties exist[26], and CsCCoAOMT1 can methylate the 6-, 7- 8-, and 3′-OH of flavonoids[27]. However, these studies did not show the activities of these enzymes on the precursor PMFs.

Rapid advances in whole-genome sequencing techniques have enabled researchers to decipher several major citrus genomes, including those of sweet orange (*Citrus sinensis*)[28], pummleo (*Citrus maxima*)[29], citron (*Citrus medica*)[29], clementine mandarin (*Citrus clementina*)[30], Mangshan wild mandarin (*Citrus reticulata*)[31], Hong Kong kumquat (*Fortunella hindsii*)[32], trifoliate orange (*Poncirus trifoliata*)[33], lemon (*Citrus limon*)[34] and round lime (*Citrus australis*)[35], providing valuable resources for understanding citrus genetic diversity and improving important citrus traits. Although PMFs are found almost exclusively in the *Citrus* genus, only limited genomic resources can be used. Because most PMFs particularly enriched in the peel of *C. sinensis* and *C. reticulata*. Moreover, the diversity and contents of PMFs in *C. reticulata* were generally greater than those in *C. sinensis*[36], suggesting that there are richer genetic resources for the biosynthesis and regulation of PMFs in the genome of CRC, one of the early cultivated varieties of *C. reticulata*, which has long been regarded as a medicinal mandarin, and is the best material for research. The lack of genomic resources for CRC prevents us to understand how PMFs, the bioactive components are produced, and has hindered progress in breeding, identification, phylgenetic evolution and disease management in CRC.

Here, we report a high-quality, chromosome-scale genome of CRC by incorporating Oxford Nanopore long-read and BGISEQ short-ead data as well as high-throughput chromatin capture Hi-C technologies. The transcriptome and metabolome data were combined to investigate the gene regulation network of PMF biosynthesis. Several putative OMT genes responsible for the *O*-methylations of PMFs were identified based on sequence similarity and domain architecture searches. Finally, a putative OMT gene responsible for HPMF (the most anti-oxidative PMF) production was characterised by in vivo and in vitro experiments. The multi-omics resources of CRC provide a foundation for future genomics-assisted breeding, medicinal fruit quality improvement and functional gene identification for the biosynthesis of plant-based medicines.

## Results
### Genome assembly and annotation
The estimated genome size of CRC was predicted to be 284.29 Mb, with a 1.01% heterozygosity ratio, based on the analysis of k-mer depth distribution (Supplementary Fig. 1). An integrated method based on the combination of Oxford Nanopore long-reads, BGI-SEQ short reads, and Hi-C data was used to assemble the genome. A primary assembly was constructed with 40 Gb of nanopore long-read data. Short reads data were subsequently used for correction and polishing, and the size of the corrective assembly was 314.96 Mb, with a contig N50 of 16.22 Mb. To obtain the chromosome-scale assembly, approximately 35 GB of Hi-C data was used. The final assembly contained 9 pseudo-chromosomes (Supplementary Data 1) with a scaffold N50 of 31.75 MB (Fig. 1b; Table 1), and 99.11% of the genome sequences were effectively anchored (Supplementary Fig. 2). To further assess the quality of the final assembly, BUSCO analysis was used to analyze completeness, and the results showed that 97.7% of the conserved genes were included, and that the assembly had only 1.10% duplication. The BGI-SEQ short reads were also aligned to the final assembly and 99.12% of the reads could be successfully mapped.

Using a combination of de novo and homology-based methods, we found that 45.04 % of the CRC genome was composed of repetitive sequences. Transposable elements (TEs) composed the majority of these repetitive sequences (43.38%, 136.63 Mb), and long terminal repeat (LTR) retrotransposons composed the largest proportion of TEs, accounting for 32.54% of the genome (Supplementary Data 2).

Multiple strategies, including ab initio, homology-based, and RNA-seq-based approaches, were used to predict protein-coding genes. A total of 29,722 genes were predicted to have an average mRNA length, coding sequence length, and exon number of 2410 bp, 1021 bp, and 4.84, respectively (Table 1). The annotation assembly resulted in 95% BUSCO completeness and 0.7% duplication. A total of 91.7% of the protein-coding genes were successfully aligned to six public databases (NR, KEGG, COG, Swiss-Prot, Interpro, and TrEMBL) with at least one hit (Supplementary Data 3). A total of 1759 transcription factors were also detected. Of these, AP2/ERF-ERF was the largest group, followed by NAC (Supplementary Data 4). The non-coding genes were also predicted, including 113 miRNAs, 549 tRNAs, 408 rRNAs, and 752 snRNAs (Supplementary Data 5).

### Phylogenetic analysis and genome evolution
To investigate the evolutionary position of CRC in the genus *Citrus*, CRC genome was compared with 9 citrus and citrus-related species, namely, *C. sinensis*, *C. clementina*, *C. reticulata*, *C. unshiu*, *C. maxima*, *C. medica*, *C. ichangensis*, *P. trifoliata*, and *Atalantia buxifolia*. Five outgroup species *Amborella trichopoda*, *Oryza sativa*, *Vitis vinifera*, *Arabidopsis thaliana*, and *Theobroma cacao* were also selected to construct the phylogenetic tree. Among the 10 *Citrus* species, a total of 13,474 gene families were shared and similar single-copy ortholog numbers were identified (Fig. 2a, b). A total of 1855 single-copy genes from the 15 representative species were used to construct the phylogenetic tree (Fig. 2c). CRC has a closer relationship to *C. clementina* than to *C. reticulata* (Mangshan wild mandarin), which is consistent with a previous study showing that *C. clementina* belongs to a modern group and Mangshan wild mandarin belongs to the group of older mandarins[37,38].

By analyzing the expansion and contraction of gene families among the 15 species, 177 expanded and 275 contracted gene families were identified in CRC. The genes associated with the expanded gene families were enriched in GO terms such as "ATPase activity, coupled to transmembrane movement of substances", "iron ion binding", and "*O*-methyltransferase activity" (Supplementary Data 6). KEGG enrichment analysis revealed that the expanded genes were related to environmental stress and secondary metabolism such as "ATP-binding cassette, subfamily C (CFTR/MRP), member 2", "heat shock 70 kDa protein 1/2/6/8", "HSP20 family protein", and "caffeic acid 3-*O*-methyltransferase" (Supplementary Data 6).

To determine the number of whole-genome duplication (WGD) events in CRC, intragenomic synteny analysis was performed,

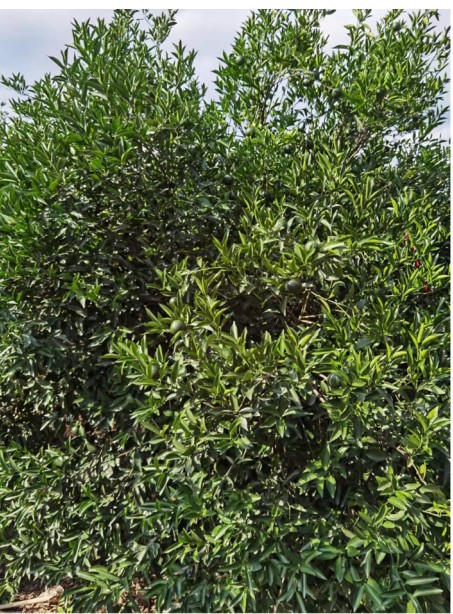

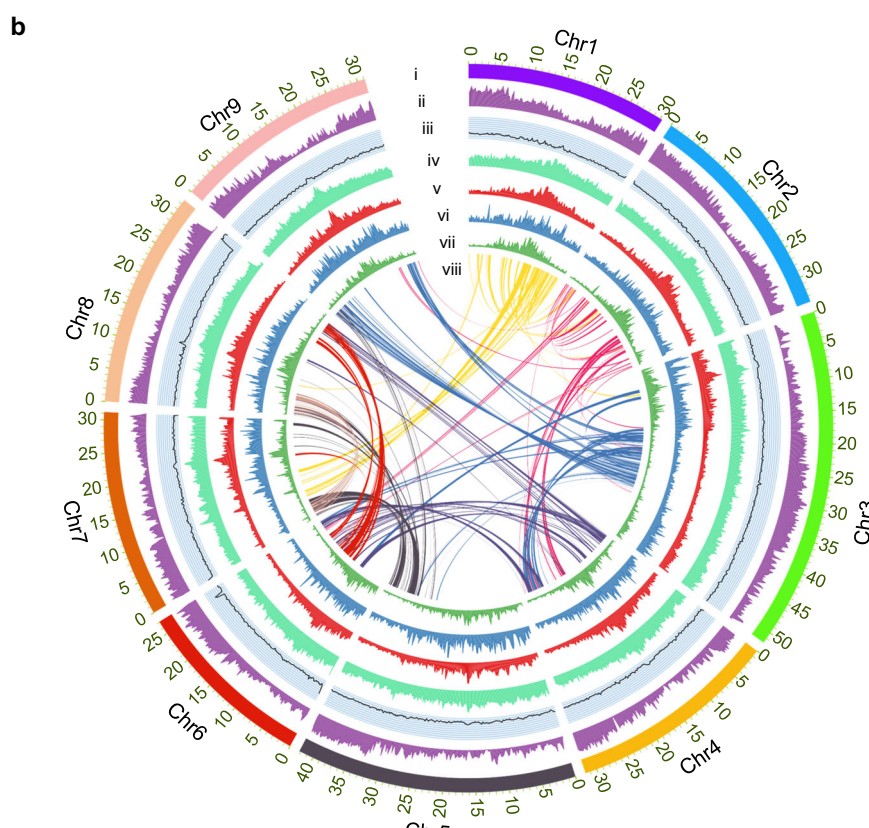

**Fig. 1 | Morphology and genome features of CRC. a** Chachi mandarin tree. **b** The landscape of CRC genome. (**i**), pseudochromosomes; (**ii**), gene density; (**iii**), GC content; (**iv**), repeat density; (**v**), LTR density; (**vi**), *Copia* density; (**vii**), *Gypsy* density; (**viii**), gene synteny.

which revealed that the genome underwent an ancient WGD event. To further investigate WGD event in *Citrus*, the number of synonymous substitutions per synonymous site (Ks) of CRC and *C. sinensis* was analyzed. The results showed a Ks peak near 1.5, which indicated that CRC only experienced a WGD event shared with all eudicot plants (Fig. 2d, e). This result was confirmed by the analysis of the four-fold synonymous third-codon transversion rate (4DTv), which is consistent with previously published results[28,33] (Supplementary Fig. 3).

According to the analysis of intragenome synteny between Chachi mandarin, Clementina mandarin, and sweet orange, there was a low frequency of chromosomal fragment rearrangement (Fig. 2f).

**Identification of key genes potentially involved in PMF biosynthesis**

PMFs are abundant in the peel of CRC. OMTs are responsible for adding *O*-methyl groups to substrates in the biosynthetic pathway of PMFs, which greatly increases the antioxidant activity of the products.

**Table 1 | Assembly and annotation of the *C. reticulata* cv. Chachiensis genome**

| Assembly | |
| --- | --- |
| Estimated genome size (Mb) | 284.29 |
| Assemble genome size (Mb) | 314.96 |
| GC content | 34.33% |
| N50 of contigs (Mb) | 16.22 |
| N50 of scaffold (Mb) | 20.33 |
| Total length of contig (Mb) | 314.95 |
| Longest scaffold (Mb) | 31.24 |
| Complete BUSCOs | 98.30% |
| Anchor size (Mb) | 312.16 |
| Anchor rate | 99.11% |
| Number of pseudochromosomes | 9 |
| N50 of scaffold (Mb, after Hi-C) | 31.76 |
| Longest scaffold (Mb, after Hi-C) | 51.75 |
| **Annotation** | |
| Number of protein-coding genes | 29,722 |
| Average length of mRNA (bp) | 2410.34 |
| Average length of CDS (bp) | 1021.2 |
| Average exon length (bp) | 210.59 |
| Avergae intron length (bp) | 361.03 |
| Average of exon number | 4.84 |
| Complete BUSCOs | 97.70% |

To identify potential OMT genes in CRC, the protein domain identification methods were used to predict functional terms of unknown proteins. A total of 47 high-confidence OMT genes were identified by sequence similarity and domain architecture searches. A total of 46.8% of these OMT genes were distributed mainly on chromosome 3 and chromosome 9, 42.5% of which were derived from tandem duplication followed by proximal duplication (21.3%). Similar to those in sweet orange, the OMT genes in CRC were unevenly dispersed across the nine chromosomes[39] (Supplementary Fig. 4), but the number of OMT genes was less than that in sweet orange. We also constructed a phylogenetic tree based on the maximum likelihood method of these confidence OMT genes from public databases, including those of citrus species, *A. thaliana*, *O. sativa*, *Nicotiana tabacum* and other plants (Fig. 3a). All OMT genes of CRC were clustered into two groups, 39 of which belonged to the COMT group, and 8 of which belong to the CCoAOMT group (Fig. 3a). Several OMT genes in CRC were clustered into a clade with known OMTs. For example, in the CCoAOMT cluster, *CZG jg3691* and *CZG jg14008* showed 93.57% and 96.79% similarity with the CCoAOMT-like gene (*CICLE_v10026344mg*), respectively, and grouped with the same branch. *CICLE_v10026344mg* from *C. clementina* has been proven to exhibit *O*-methyltransferase activity toward 6-OH-, 8-OH- and 3′-OH- containing compounds[25]. In addition, *CZG jg17822* was grouped with *CitOMT* (BBU25484.1) in the COMT cluster. *CZG jg17822* showed 99.73% similarity with *CitOMT*, which has been reported to catalyze the transfer of a methyl group to 3′-OH[26], implying that the unknow COMT gene might have similar functions in CRC.

To decipher the expression pattern of OMT genes in CRC, transcriptomic data from flowers to mature fruits were analyzed (Fig. 3b). Among these OMT genes, most CCoAOMT genes were highly expressed in young fruits and peels followed by flowers and leaves, and their expression in seeds and pulp could hardly be detected (Fig. 3c). Furthermore, the expression profile of the CCoAOMT genes did not significant differ in different developmental stages. In contrast to the

CCoAOMT genes, 24 of 39 COMT genes were expressed in most of the tissues and could be clustered into five groups based on the similar gene expression patterns (Fig. 3d), while the remaining COMT genes exhibited relatively low expression levels. The COMT genes in group I exhibited high expression levels in young tissues, such as young fruits (45 days after flowering, DAF and 75 DAF), flowers, and tender leaves. The genes in group II were highly expressed in young fruits and peels but not in leaves, pulp, or seed, but group III had relatively high expression in nearly mature (165 DAF and 200 DAF) and mature peels (230 DAF and 260 DAF) as well as in leaves. Interestingly, unlike the above groups with relatively low expression levels in the pulp and seeds, group IV and group V showed the opposite patterns. One of the genes, *CZG jg17822*, had a markedly higher expression level in seeds and pulp of all developmental stages followed by the peel (Fig. 3d and Supplementary Fig. 5). The expression pattern of COMT genes in CRC varied across the samples, suggesting that the activity of COMT changed during different developmental stages, which might contribute to the differences in the accumulation of PMFs.

## The metabolome profile of PMFs from CRC at different developmental stages

To profile the flavonoid metabolome of CRC, fruits or peels from five different developmental stages, including young fruits at 45 DAF and 75 DAF, unmatured peels at 105 DAF, nearly mature peels at 200 DAF and mature peels at 260 DAF, were collected. A total of 304 flavonoid metabolites were detected in the fruits and peels using UPLC–MS/MS (Supplementary Fig. 6; Supplementary Data 7). Among these, 29 PMFs were detected in all the samples (Fig. 4; Supplementary Data 8; Supplementary Figs. 7–18).

A comparison between 45 DAF and 75 DAF revealed significant differences in the relative content of flavonoid compounds. Among the 103 differentially accumulated metabolites (DMs, VIP ≥ 1 and *P* < 0.05), 8 PMFs had relatively high content in 75 DAF (Supplementary Data 9), including 5,7,4′-trihydroxy-3,6,3′,5′-tetramethoxyflavone, natsudaidain, natsudaidain-3-*O*-(5′-glucosyl-3-hydroxy-3-methylglutarate)glucoside, 5,7,8,3′,4′-pentamethoxyflavanone, 2′-hydroxy-3,4,5,5′,6′-pentamethoxychalcone, HPMF, 5-hydroxy-6,7,3′,4′-tetramethoxyflavanone and natsudaidain-3-*O*-(3-hydroxy-3-methylglutarate)glucoside. The relative content of natsudaidain-3-*O*-(5′-glucosyl-3-hydroxy-3-methylglutarate)glucoside showed the largest change (fold change > 9.56) between the two periods, followed by natsudaidain-3-*O*-(3-hydroxy-3-methylglutarate)glucoside and 5,7,8,3′,4′-pentamethoxyflavanone (fold change > 4.88 and fold change > 4.43, respectively). In contrast, 5,6,7,8,3′,4′-hexamethoxyflavanones was abundant in 45 DAF. In addition, naringenin, the precursors of the major flavonoid compounds, was also more highly accumulated at 45 DAF.

From unmatured to mature stage, we performed three different comparisons, including 105 DAF vs 200 DAF, 105 DAF vs 260 DAF and 200 DAF vs 260 DAF (Supplementary Data 10). Five PMFs were DMs (VIP ≥ 1, *P* < 0.05) between 105 DAF and 200 DAF, 2 of which were abundant at 105 DAF including natsudaidain-3-*O*-(5′-glucosyl-3-hydroxy-3-methylglutarate)glucoside, and natsudaidain-3-*O*-(3-hydroxy-3-methylglutarate)glucoside. The other three DMs 5,6,7,8,3′,4′-hexamethoxyflavanone, 5,7,3′,4′,5′-pentamethoxyflavanone and 5,7-dihydroxy-6,3′,4′,5′-tetramethoxyflavone (arteanoflavone) were more abundant at 200 DAF.

In the comparison between 200 DAF and 260 DAF, 6 PMFs were differentially accumulated (VIP ≥ 1, *P* < 0.05), all of which showed increased abundance at 200 DAF, including 5,6,7,4′-tetramethoxyflavanone, 5,7-dihydroxy-6,3′,4′,5′-tetramethoxyflavone, 5,7,3′,4′,5′-pentamethoxyflavanone, 5,7,4′-trihydroxy-3,6,3′,5′-tetramethoxyflavone, natsudaidain and natsudaidain-3-*O*-(3-hydroxy-3-methylglutarate)glucoside.

 

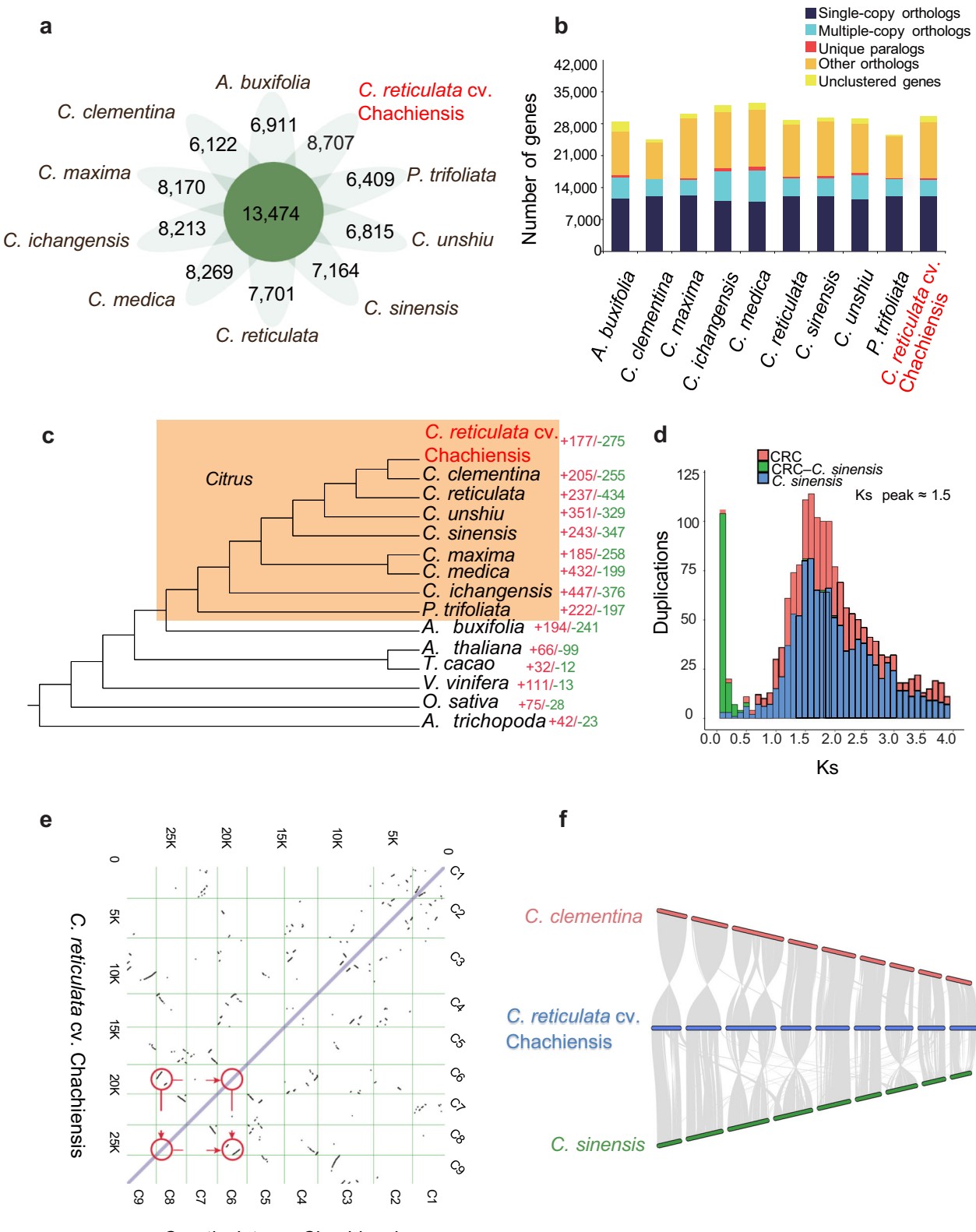

**Fig. 2 | Comparative genomic analysis of CRC and other species. a** Floral diagram of shared gene families. **b** Gene family number distribution in 10 citrus species. **c** Phylogenetic tree of 15 plant species. **d** Ks distribution of WGD-derived gene pairs in Chachi mandarin and sweet orange. **e** Dot plot of paralogs in the genome of CRC, the red circles indicate the ancient WGD event. **f** Genome synteny analysis between CRC, *C. sinensis*, and *C. clementina*.

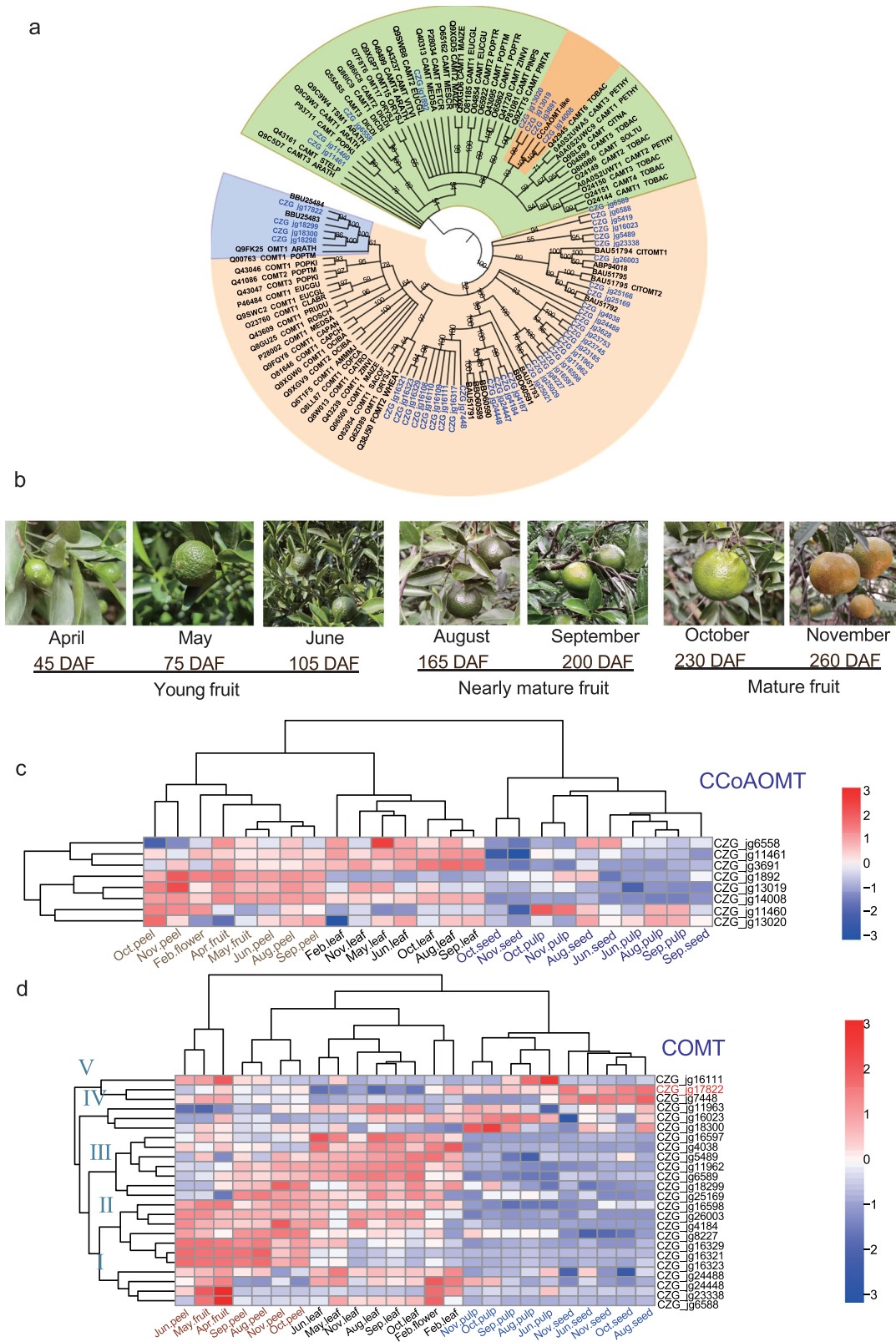

**Fig. 3 | OMT genes in CRC. a** Phylogenetic tree of OMT in *Citrus* and other plants. CCoAOMT genes are colored in green, and COMT genes are colored in fleshcolor. The blue and orange branches indicate Chachi mandarin OMT genes that are clustered with known OMTs, and all OMTs in CRC are indicated in blue. **b** Different developmental stages of CRC fruits. **c** Expression profile of CCoAOMT among different developmental stages and different tissues. **d** Expression profile of COMT among different developmental stages and different tissues.

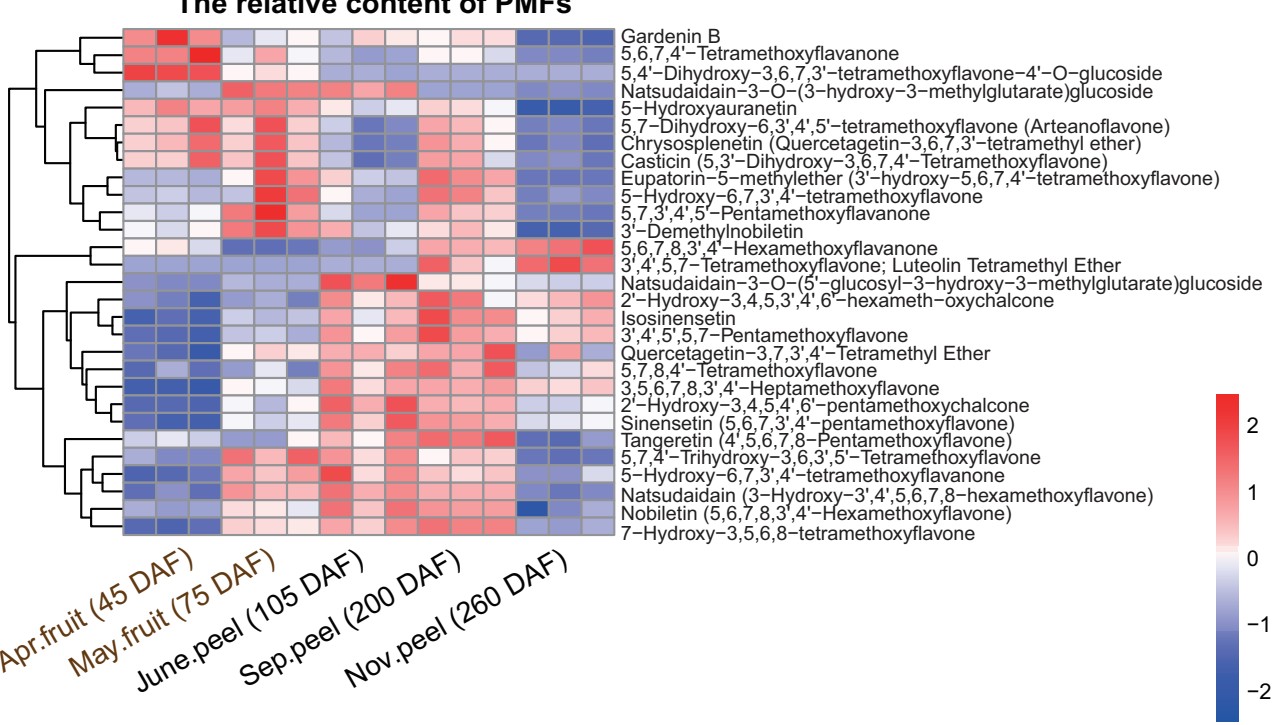

**Fig. 4 | The relative content of 29 PMFs in CRC.**

Interestingly, the comparison between 105 DAF and 260 DAF showed that 3′,4′,5,7-tetramethoxyflavone and 5,6,7,8,3′,4′-hexamethoxyflavanone were abundant at 260 DAF (VIP ≥ 1, $P < 0.05$), while the relative content of 4 PMFs, including 5,7,4′-trihydroxy-3,6,3′,5′-tetramethoxyflavone, natsudaidain, natsudaidain-3-O-(3-hydroxy-3-methylglutarate)glucoside and natsudaidain-3-O-(5′-glucosyl-3-hydroxy-3-methylglutarate)glucoside, decreased at 260 DAF.

A total of nine significantly differentially enriched PMFs among the three developmental stages were identified, seven of which were abundant at 200 DAF. The accumulation of 13 PMFs also reached their highest relative content at 200 DAF, although the relative content of the other PMFs, such as nobiletin and sinensetin did not significantly vary among the three developmental stages (Supplementary Fig. 19). These results indicated that 200 DAF might be the best stage to harvest the peel of CRC.

Moreover, the dynamic changes in the relative content of natsudaidain and its derivatives natsudaidain-3-O-(3-hydroxy-3-methylglutarate) glucoside and natsudaidain-3-O-(5′-glucosyl-3-hydroxy-3-methylglutarate)glucoside showed significantly strong correlations (r = 0.54, and r = 0.74, respectively) at different developmental stages, indicating that the 3-OH of natsudaidain might be the active site. There was also a strong correlation between the relative content of natsudaidain and HPMF (r = 0.628, $P = 0.012$), indicating the possible production of HPMF via natsudaidain. HPMF is an important medicinal component with anticancer activities[32], however, the chemical reaction of methyl transfer is still unclear. According to the molecular structure of HPMF, natsudaidain could also be transformed into this compound by transferring a methyl group to the 3-OH position of natsudaidain.

### CcOMT1, a candidate enzyme involved in the biosynthesis of HPMF

We hypothesized that HPMF might be transformed from the precursor compound natsudaidain, by the OMT which transfers a methyl group to the 3-OH site. Then, we used the sequence of ShMOMT3, a 3-OMT of flavonoids in tomatoes[40], to search against the assembled-protein database of CRC. Several highly similar candidate genes were obtained from the BLAST search including *CZG_jg11962* (52%), *CZG_jg11963* (51%), *CZG_jg17822* (51%), *CZG_jg18299* (51%), *CZG_jg23745* (50%) and *CZG_jg23753* (50%). Gene-specific primers were designed to obtain candidate genes from the cDNA library of CRC leaves. All these candidate genes were subsequently cloned and tested.

Interestingly, only the protein encoded by the gene *CZG_jg17822* which was predicted to catalyze O-methylation on 3′-position displayed the expected function (catalyze O-methylation on 3-hydroxyl group) and was named CcOMT1. The open reading frame (ORF) of *CcOMT1* was 1080 bp and encoded a putative protein of 366 amino acids. The calculated isoelectric point and the predicted molecular weight of the amino acid sequence were 5.70 and 39.96 kDa, respectively (ExPASy Compute pI/Mw Tool).

To test the enzyme activity of CcOMT1 in vitro, the recombinant plasmid pET-28a (+)-CcOMT1 was transformed into *E. coli* BL21(DE3). After induction and purification, a single band of 39.96 kDa was obtained by SDS−PAGE analysis, which is consistent with the theoretically calculated molecular weight of the recombinant protein (Supplementary Fig. 20). The optimal pH and temperature of CcOMT1 were measured using natsudaidain as the substrate. The highest reaction rates were observed at 55 °C and pH 6.0 in the $NaH_2PO_4$-$Na_2HPO_4$ buffer (Supplementary Fig. 21). Further experiments were performed at 50 °C and pH 6.0 to balance enzyme activity and substrate stability. The reactions were analyzed by HPLC/DAD and LC-QTOF−MS. The results showed that a new peak appeared on the chromatogram (Fig. 5b). Compared with the reference standards, the methyl-product displayed abundant [M + H]+ ions at m/z 433 of **1a** in MS/MS, which was identified as HPMF (Fig. 5c). These results indicated that CcOMT1 can catalyze the 3-OH methylation of natsudaidain. In addition, we also conducted a set of experiments to examine the promiscuity of CcOMT1, which indicated that CcOMT1 could catalyze not only the 3-hydroxyl group of flavonoids, but also the 5-, 7-, 3′-, 4′- hydroxyl groups of flavonoids (Supplementary Fig. 22). Nevertheless, this result confirmed the above prediction that CcOMT1 (CZG_jg17822) might methylate the 3′-hydroxyl group of flavonoids.

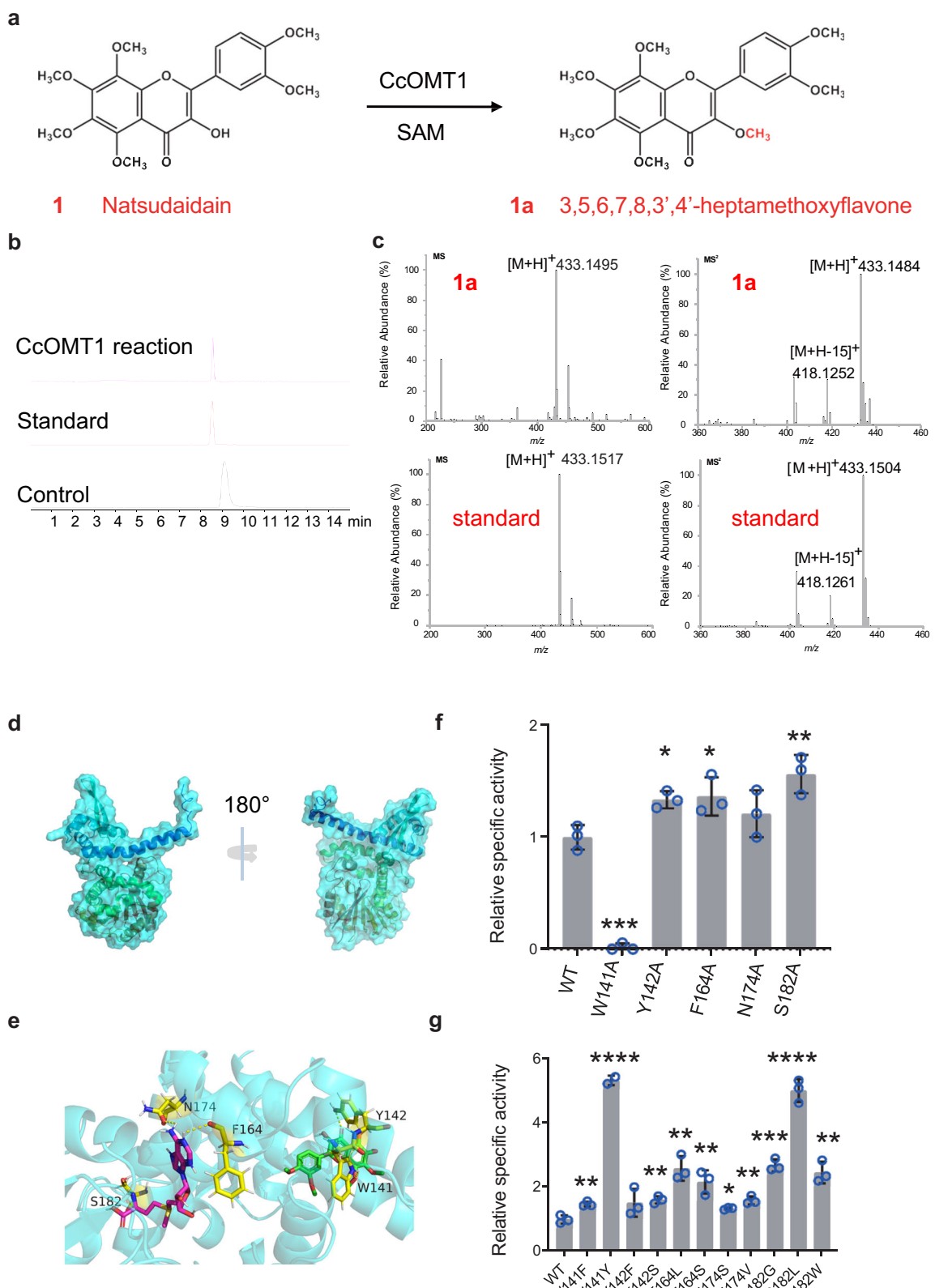

**Fig. 5 | Catalytic function and mutants of CcOMT1. a** Methylation of natsudaidain (**1**) catalyzed by CcOMT1. **b** Total ion current chromatograms for the enzymatic reactions. The reaction mixtures were incubated at 50 °C for 4 h. **c** Tandem mass spectra of product **1a**. **d** Structure model of CcOMT1. SAM and natsudaidain are shown in magenta and green, respectively. **e** Molecular docking with natsudaidain into the substrate binding pocket of CcOMT1. **f** Relative activities of alanine-substituted mutants of CcOMT1. **g** Effects of different mutants on enzyme reactivity. The error bars represents the standard deviation from three replicates.

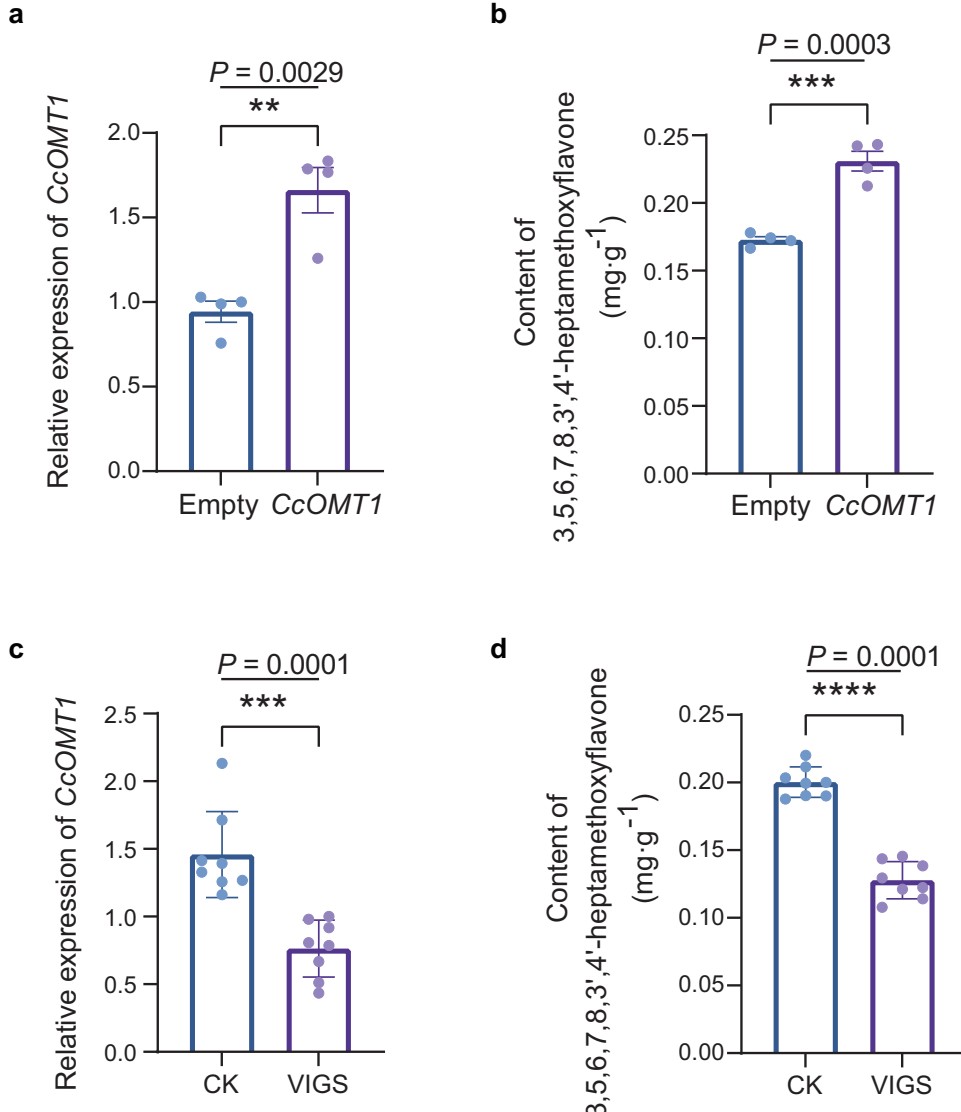

**Fig. 6 | Transient overexpression, silencing and functional analysis of *CcOMT1*.** **a** Expression profiles of *CcOMT1* after infiltration with CcOMT1-pCAMBIA1301. **b** Changes in 3,5,6,7,8,3',4'-heptamethoxyflavone (HPMF) content after transient overexpression of *CcOMT1* in CRC peel. The effect of transient expression of *CcOMT1* on the content of HPMF in the peel of CRC was measured after infiltration with CcOMT1-PCAMBIA1301, with the empty vector as a control. Error bars, mean ± s.d, *n* = 4. The experiment was repeated twice. *** indicates a significant difference between the two groups according to a one-tailed Student's *t* test. **c** Relative expression of *CcOMT1* in virus-induced *CcOMT1* silencing CRC fruits. **d** Changes in HPMF content in CRC fruits after virus-induced silencing of *CcOMT1*. CK, empty TRV2 vector control fruits; VIGS, virus-induced *CcOMT1* gene silencing fruits. The content of flavonoids is expressed as mg·g[1] FW. Error bars, mean ± s.d, *n* = 8. The experiment was repeated twice. *** indicates a significant difference between the two groups according to a one-tailed Student's t test.

To determine the kinetic parameters of CcOMT1, the Michaelis–Menten constant (Km) was determined via non-linear regression fitting, and the turnover number (Kcat) was calculated (Supplementary Fig. 23). We also compared the activity of several OMTs that catalyze flavonoids (Supplementary Data 11). The Kcat/Km value indicated that CcOMT1 had low catalytic activity. Therefore, site-specific mutation based on semi-rational design was performed on CcOMT1 to improve its catalytic activity.

Semi-rational design guided by structural information is an effective strategy for improving catalytic activity. To determine the suitable target residue for semi-rational mutagenesis, a homology model of CcOMT1 was constructed through Robetta (Fig. 5d)[41]. The donor SAM was docked into the substrate acceptor pocket using AutoDock vina version 1.5.6.

Alanine scanning can be used to detect the contribution of individual amino acid side chains to enzyme function[42]. Five residues (W141, Y142, F164, N174 and S182) within 5 Å of substrate **1** and methyl donor SAM binding site were selected and mutated into alanine by site-directed mutagenesis (Fig. 5e). The activity of these mutants was tested in vitro. The activities of four of the five mutants (Y142A, F164A, N174A and S182A) were greater than that of the wild type (WT). However, the activity of the W141A mutant was quite low (< 10%) (Fig. 5f).

To further enhance the activity of CcOMT1, five residues (W141, Y142, F164, N174, and S182) were mutated according to their hydrophobic side chains and molecular size. Then, 11 mutants (W141F, W141Y, Y142F, Y142S, F164S, F164L, N174V, N174S, S182L, S182G, and S182W) were obtained, and their enzymatic activities were determined. All the mutants showed increased activity, with W141Y and S182L exhitbiting around 4-fold increased activity relative to the WT (Fig. 5g). Therefore, the results indicated that W141 and S182 are important amino acid residues that affect the reactivity of CcOMT1.

To further assess the potential role of CcOMT1 in the methylation of flavonoids in vivo, transient overexpression experiments were conducted. The transcript levels of *CcOMT1* and the content of HPMF in CRC peel infiltrated with *CcOMT1*-pCAMBIA1301 were both increased significantly compared to the empty vector (Fig. 6a, b). The virus-induced gene silencing (VIGS) system was also used to test the function of *CcOMT1*. *Agrobacterium tumefaciens* cultures with TRV1/TRV2 or TRV1/TRV2-*CcOMT1* constructs were transiently infiltrated into CRC fruits. Compared with the empty vector (TRV1/TRV2), VIGS significantly reduced the expression level of *CcOMT1* and the accumulation level of HPMF (Fig. 6c, d). The results of transient overexpression and silencing of *CcOMT1* suggested that CcOMT1 might be involved in the biosynthesis of HPMF in CRC.

However, the enzymatic activity of both CcOMT1 and its mutants on natsudaidain is low, which may be related to the low content of the target compound HPMF in CRC. A recent study reported that the content of HPMF is as low as 0.75 mg·g$^{-1}$ DW at 60 days post anthesis in the fruit flavedo of CRC, while the content of the precursors of HPMF, such as nobiletin and tangerine, are 20 mg·g$^{-1}$ DW and 11 mg·g$^{-1}$ DW, respectively[43]. The content of HPMF is only 3.75% of nobiletin and 6.82% of tangerine. Our in vivo experimental results also demonstrated that the content of HMPF in the control groups is low, approximately 0.20 mg·g$^{-1}$ FW in the fruits of CRC (Fig. 6b, d). It is suggested that HPMF is a trace ingredient in CRC, indicating that the *O*-methyltransferases related to HPMF biosynthesis should exhibit low catalytic activity toward the precursor compounds of HPMF in vivo. Therefore, we assumed that CcOMT1 is a candidate enzyme involved in HPMF biosynthesis.

### Glycosylation of PMFs and other flavonoids in CRC

Glycosylation is catalyzed by UDP-glycosyltransferase (UGT), which is pivotal for enriching the diversity of flavonoids. Several glycosylated PMFs, such as natsudaidain-3-*O*-(5′-glucosyl-3-hydroxy-3 methylglutarate)glucoside, natsudaidain-3-*O*-(3-hydroxy-3methylglutarate)glucoside and 5,4′-dihydroxy-3,6,7,3′-tetramethoxyflavone-4′-*O*-glucoside were identified in CRC fruits by UPLC-MS/MS. In addition, the relative content of natsudaidain and its glycoside derivatives (natsudaidain-3-*O*-(5′-glucosyl-3-hydroxy-3-methylglutarate)glucoside and natsudaidain-3-*O*-(3-hydroxy-3-methylglutarate)glucoside) were highly correlated, indicating the presence of glycosylation of the PMFs. PMFs might be metabolized into *O*-glycoside derivatives and stored stably in the peel. UGT might play a key role in this process. Therefore, we investigated the potential UGT genes in CRC, especially the UGT genes correlated with PMFs. First, 96 candidate genes were identified by searching for UGT genes in the genome. Then, a phylogenetic tree was constructed to determine the phylogenetic relationship of the candidate UGT genes combined with known UGT sequences extracted from *A. thaliana, C. sinensis, Zea mays, Catharanthus roseus, Pueraria lobata, Punica granatum, Ginkgo biloba,* and *O. sativa* (Supplementary Fig. 24). All UGT genes were clustered into 19 groups (Group A to R), and correlation analysis revealed that the expression levels of 78 candidate UGT genes were significantly correlated with the relative content of 29 PMFs (adjusted *P* < 0.05). Among these genes, *CZG jg11190* (Group A) had a greater correlation with natsudaidain and its derivative natsudaidain-3-*O*-(3-hydroxy-3-methylglutarate) glucoside, which might be a gene involved in natsudaidain glycosylation (cor = 0.88, cor = 0.75, *P* < 0.05, respectively).

*CZG jg16074* was grouped with *Cs1,6RhaT*, which encodes a 1,6-rhamnosytransferase that produces the tasteless flavanone-7-*O*-rutinosides in citrus[44], indicating that *CZG jg16074* might underlie the biosynthesis of the non-bitter flavanone in CRC. We also detected glycosylated flavanones with a rutinoside, such as naringenin-7-*O*-rutinoside-4′-*O*-glucoside, naringenin-7-*O*-rutinoside (narirutin), hesperetin-7-*O*-rutinoside (hesperidin), eriodictyol-7-*O*-rutinoside (eriocitrin), diosmetin-7-*O*-rutinoside (diosmin), and didymin

(isosakuranetin-7-*O*-rutinoside) in our samples (Supplementary Data 7). Interestingly, *Crc1,6RhaT* which was recently cloned from CRC showed 99.86% sequence similarity with *CZG_jg16074*, was identified as a hesperidin synthase gene, and Crc1,6RhaT can transform flavanone-7-*O*-glucoside into hesperidin[45], indicating that CZG_jg16074 might have a similar function.

In group R, the candidate flavonoid *C*-glucosyltransferase (CGT) gene *CZG jg8991* was identified, and showed high similarity with the known *CGT* (UGT708C2). UGT708C2 is a 2-hydroxyflavanone *C*-glycosyltransferase that shows glucosylation toward 2-hydroxynaringenin and generates the corresponding compounds vitexin and isovitexin[46]. These two compounds were also detected in our samples (Supplementary Data 7), and the candidate CGT gene exhibited relatively high expression levels in pulp and peel, which indicated that *CZG jg8991* might contribute to a greater content of the corresponding products in CRC fruits (Supplementary Fig. 25).

### A possible gene regulation network of PMF biosynthesis

Most of the PMFs exhibited the highest relative content in the peel of CRC at 200 DAF, followed by 105 DAF, suggesting that the gene expression and regulation of PMF biosynthesis could be active during these stages. A total of 4604 differentially expressed genes (DEGs) were identified between 105 DAF and 200 DAF (|log2foldchange| >1, adjusted *P* < 0.05), and 75 DMs (VIP ≥ 1 and fold change > 2) were also be identified during this period. GO enrichment analysis revealed that most of the DEGs were enriched in the pathway of "transcription factor activity, sequence-specific DNA binding" (Supplementary Fig. 26a). KEGG enrichment analysis revealed that most of the DEGs were enriched in the plant hormone signal transduction pathway followed by the "MAPK signaling pathway – plant" pathway (Supplementary Fig. 26b). These results indicated that plant hormones, environmental stress and transcriptional regulation might be related to flavonoid accumulation in CRC.

To decipher the gene regulation network of the PMF biosynthetic pathway, a correlation network of the differentially expressed transcription factors, flavonoid biosynthesis–related genes and the corresponding 75 DMs was constructed. The transcription factors, such as AP2/ERF family, WRKY, and bZIP, with the most degree showed significant correlations with flavonoid biosynthesis genes, such as chalcone isomerase (CHI) gene, flavonol synthase (FLS) gene, and flavanone-3-hydroxylase (F3H) gene (Supplementary Fig. 27; Supplementary Data 12). These findings suggested that these transcription factors might regulate the expression level of flavonoid biosynthesis genes. AP2/ERF-ERF transcription factors also showed a significant correlation with 5,6,7,8,3′,4′-hexamethoxyflavone (Supplementary Data 13). Therefore, we hypothesized that the transcription factor AP2/ERF family might regulate the accumulation of PMFs.

To further verify our hypothesis, the weighted gene co-expression network analysis among the five developmental stages samples (45 DAF-fruit, 75 DAF-fruit, 105 DAF-peel, 200 DAF-peel and 260 DAF-peel) was performed, generating 18 co-expression modules. The MEyellow module was strongly correlated with the 200 DAF near mature peel (r = 0.89, *P* = 9e-06) (Supplementary Fig. 28) and significantly positively correlated with PMFs (r = 0.7, *P* = 0.004) and flavonoid *C*-glycoside (r = 0.8, *P* = 3e-04) (Supplementary Fig. 29). The MEblue module was correlated with the 45 DAF fruit (Supplementary Fig. 28) and negatively correlated with flavonols (r = -0.7, *P* = 0.004) and flavonoid-*O*-glucosides (r = -0.67, *P* = 0.006; Supplementary Fig. 29). In the co-expression network of the two modules (Fig. 7a), AP2-ERF/ERF transcription factors were identified as hub genes and strongly associated with flavonoid biosynthesis-related genes such as *FLS* and *UGT*, indicating that AP2-ERF/ERF transcription factors might play an important role in modulating the flavonoid biosynthesis. Other transcription factors, such as LIM, SET, and MYB, also showed associations with the flavonoid biosynthesis genes in the MEblue module (Fig. 7b). In

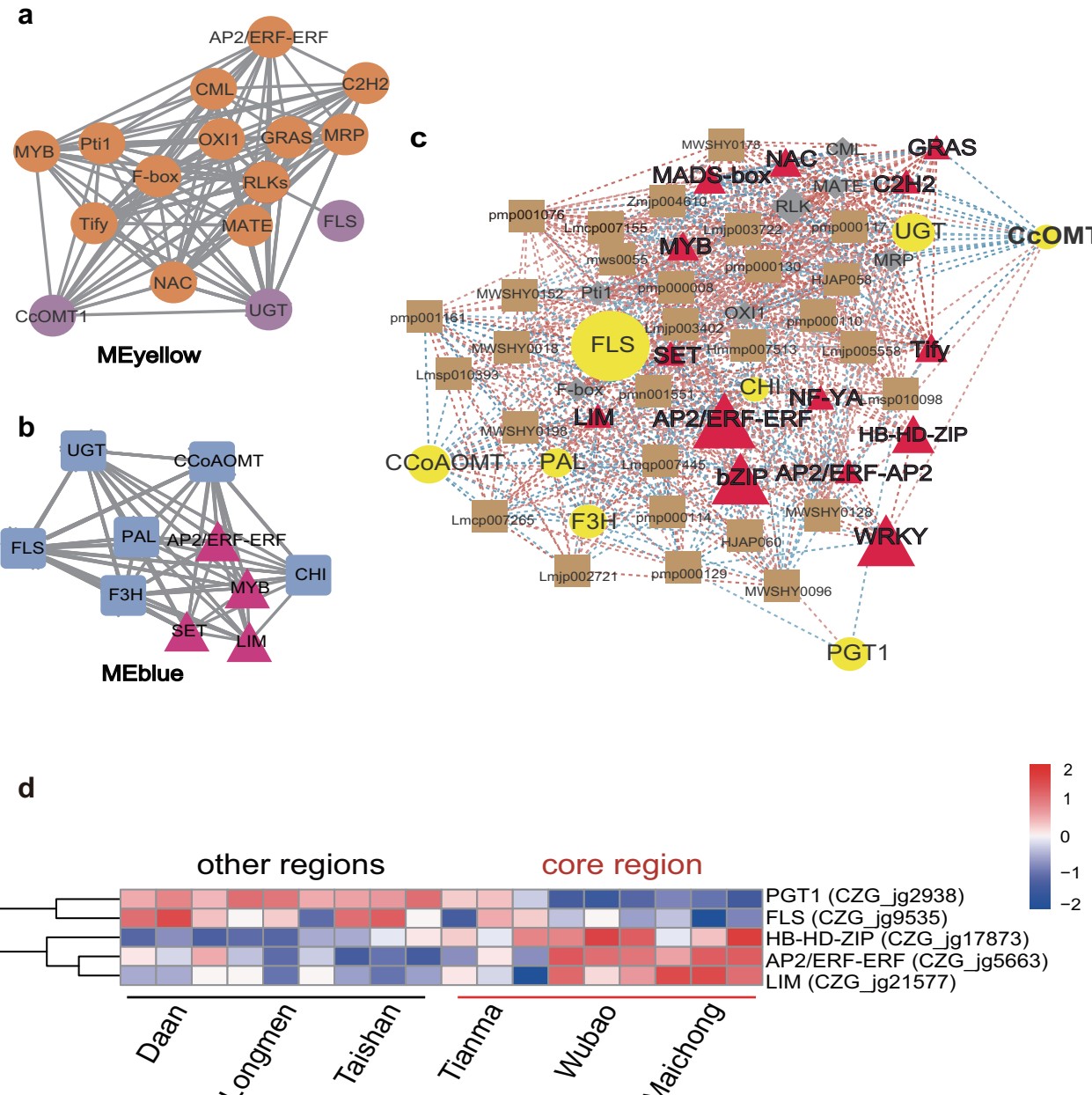

**Fig. 7 | The potential gene regulation network of PMF biosynthesis. a** The co-expression network in the yellow module. The orange circles indicate hub genes and the purple circles indicate PMF biosynthesis genes. **b** The co-expression network in the blue module. Triangles indicate hub genes and squares indicate PMF biosynthesis related genes. **c** The correlation network of the hub genes and PMFs. Red triangles indicate transcription factors; gray diamonds indicate stress related proteins; yellow circles indicate PMF biosynthesis related genes; brown squares represent PMFs; red dashed lines indicate the positive correlation and blue dashed lines indicate the negative correlation. PAL phenylalanine ammonia-lyase, CHS chalcone synthase, CHI chalcone isomerase, CCoAOMT caffeoyl-CoA *O*-methyltransferase, F3H flavanone-3-hydroxylase, FLS flavonol synthase, UGT UDP-glycosyltransferase, PGT1 phlorizin synthase. **d** The expression profile of potential regulator genes between the core region and other regions.

addition, the transcription factors AP2/ERF-ERF, MYB, Tify, and NAC showed associations with *CcOMT1* in the MEyellow module, implying that the functions of transcription factors might be involved in regulating the *O*-methylation of PMFs. Besides, some environmental stress-related proteins, such as Pti1[47] and F-box[48], were also associated with *CcOMT1*, which might indicate that the production of PMFs is related to the stress response.

We further merged the correlation network and co-expression network by calculating the Pearson coefficient of all flavonoid biosynthesis-related genes, transcription factors and PMFs (Fig. 7c). Transcription factors, such as AP2/ERF-ERF, bZIP, and WRKY, as well as LIM, might interact with the biosynthetic genes such as *CHI* and *FLS*. AP2/ERF-ERFs were identified as key genes in the potential regulatory network, which might drive related gene expression and promote the accumulation of PMFs. However, how AP2/ERF-ERF works in CRC needs further confirmation.

### Differential expression of the potential key regulatory genes between the core region and other regions
Xinhui District, Jiangmen City of Guangdong Province, is considered the core region (geo-authentic product region) for producing the traditional medicine "Guangchenpi". Although, CRC is planted in other

cities of Guangdong and Guangxi Province, the quality of citrus peels varies between the core region and other regions. Therefore, we investigated whether the expression levels of the potential key genes in the regulatory network significantly differed between the core region and other regions. The transcriptomic data of CRC peels from three representative orchards (Tianma, Wubao and Maichong) in the core region and three orchards (Taishan, Longmen, and Daan) far from the core region were analyzed. The results showed that AP2/ERF-ERF (*CZG jg5663*), LIM (*CZG jg21577*), and HB-HD-ZIP (*CZG jg17873*) exhibited relatively high expression levels in the core region, possibly contributing to the difference in CRC peel between the core region and other regions (Fig. 7d). This result partially suggested the potential roles of AP2/ERF-ERF, LIM, and HB-HD-ZIP in regulating PMF biosynthesis.

### The proposed biosynthetic pathway of PMF in CRC

All PMFs are derived from the main flavonoid naringenin (Fig. 8). Early flavonoid biosynthesis genes such as *CHS* and *CHI* contributed to naringenin production. Late flavonoid biosynthesis genes such as *FLS* and *F3H* are responsible for the modification of naringenin, resulting in the production of different types of naringenin derivatives. These chemical products could further undergo glycosylation and methylation, where UGT and OMT play essential roles, respectively. After that, PMFs with different moieties could be produced. Transcription factors such as AP2/ERF-ERF, HB-HD-ZIP, LIM, and WRKY might be involved in this process.

In the proposed pathway, natsudaidain might come from three different sources of main flavonoids, such as apigenin, luteolin and kaempferol, based on the structure of natsudaidain. Some PMFs, such as 5,7,8,4′-tetramethoxyflavone and 5-hydroxy-6,7,3′,4-tetramethoxy-flavone detected in CRC samples (Supplementary Data 7) might serve as intermediate compounds. When new hydroxyl groups are added and undergo methylation, these intermediate compounds are ultimately converted into natsudaidain. With the catalysis of CcOMT1, the most methoxylated PMF, HPMF was produced from natsudaidain. During this process, unknown flavone 6-hydroxylase (F6H) and flavone 8-hydroxylase (F8H) affect the structures of PMFs. In addition, due to the promiscuity of CcOMT1, nobiletin might be produced from 3′-demethylnobiletin by CcOMT1, which can catalyze the 5-, 7-, 3′-, 4′-hydroxyl groups and then be converted into natsudaidain with the activity of an unknown F3H (Fig. 8). CcOMT1 exhibited relatively high and stable gene expression levels throughout developmental stages, and its substrate promiscuity has made it a candidate enzyme for PMF biosynthesis in CRC.

## Discussion

There is a lack of multi-omics resources for the important traditional Chinese medicine "Guangchenpi", which is a major obstacle to studying its medicinal components at the gene level[49]. In this study, high-quality genome, transcriptome, and metabolome data of CRC were generated, which not only extends our knowledge of cultivated mandarins, but also fills the gap in knowledge of the metabolic pathways related to the medicinal compounds of "Guangchenpi". Comparative genomics analysis also showed that CRC more closely related to *C. clementina* than to *C. reticulata* (Mangshan wild mandarin), which provides evidence for future studies on citrus domestication.

PMFs are one of the most critical bioactive ingredients of "Guangchenpi". Methylation of hydroxyl groups is important for the bioactive activities of PMFs. However, to date, the number of identified functional OMT genes is very limited. In this study, 47 high-confidence OMT genes with potential functions of *O*-methylation were identified based on sequence similarity and domain architecture searches. The expression patterns of COMT genes of CRC vary among different developmental stages of fruits, which might contribute to the differences in the accumulation of PMFs. In addition, 96 UGTs were

identified, some of which might have the potential to metabolize PMFs into the storage form.

Most of the PMFs were abundant in the peel at 200 DAF compared to 105 DAF and 260 DAF, indicating that 200 DAF might be the best stage for harvesting the peel of CRC.

To elucidate the biosynthetic pathway of PMFs, we characterized CcOMT1, which can directly transfer a methyl group to the 3-OH site of natsudaidain, and produce HPMF, the most effective potential anti-oxidant and anticancer[50] metabolite in "Guangchenpi". In this study, we showed that CcOMT1 was able to catalyze directly on the PMF substrate which is different from previous works[25–27]. W141 and S182 are important amino acid residues of CcOMT1 that affect its reactivity. We further improve the natural enzyme to increase its efficiency by site-directed mutagenesis. The function of CcOMT1 was also suggested in vivo by transient overexpression and virus-induced gene silencing experiments, and it is believed to a candidate enzyme for HPMF biosynthesis. Despite the low activity of CcOMT1 in HMPF production, which may be related to the extremely low content of the product in CRC, the discovery of CcOMT1 advanced our understanding of bioactive product biosynthesis in 'Guangchenpi' at the molecular level.

We used transcriptome and metabolome data to decipher a possible gene regulatory network associated with PMF biosynthesis. The transcription factors AP2/ERF, bZIP, and WRKY family members, which have been reported to be involved in both abiotic and biotic stress[51–53] might be involved in regulating the production of flavonoids. AP2/ERF family transcription factors participate in plant responses to various abiotic stress, such as drought and salt[54]. AP2/ERF also functions in the flavonoid accumulation, for example, CitERF32 and CitERF33 positively activate the transcription and enhance the flavonoid biosynthesis in the mandarin cultivar Ougan[55]. MdAP2-34 can modulate flavonoid accumulation by enhancing *MdF3′H* promoter activity in apple[56]. Similarly, a bZIP gene has been proven to regulate flavonoid biosynthesis in grape[57]. The expression levels of regulating genes such as *AP2/ERF-ERF*, *LIM*, and *HB-HD-ZIP* were greater in the Xinhui region than in other regions. Similarly, more flavonoids accumulated in the peel of CRC from Xinhui region than those from other regions[58]. These findings suggested that increased expression of *AP2/ERF-ERF*, *LIM*, and *HB-HD-ZIP* might contribute to increased abundance of flavonoids. In the possible regulatory network, transcription factors AP2/ERFs, LIM and HB-HD-ZIP have emerged as potential key regulators of abiotic stress, such as salinity stress[59–61]. High salt stress actually occurs in Xinhui, which is supported by the natural environment. Xinhui is located at the intersection of Xijiang river and Tanjiang river where saltwater intrusion occurs every year. High salinity stress might increase the expression levels of regulatory genes and thus contribute to the high quality of "Guangchenpi".

Overall, we elucidated the evolutionary relationships of CRC in the genus *Citrus* and characterized the dynamic changes of PMF abundance at different developmental stages. We identified CcOMT1 as a candidate enzyme involved in HPMF production and propose a possible gene regulatory network of PMFs, which provide insights into "Guangchenpi" quality improvement. The multi-omics data also extended our knowledge on the PMF biosynthetic pathway in CRC. These resources and our discoveries will greatly promote CRC breeding, varieties identification and utilization.

## Methods

### Genome sequencing

Xinhui of Guangdong Province is the major and traditional growing area of CRC. Dry and mature peel from Xinhui District is known to be high quality than those from other regions[62]. All plant materials in this study were collected from a 10-year-old CRC tree in an orchard located in Xinhui. The genomic DNA of the fresh leaves of CRC was extracted using the CTAB method[63] and used for nanopore sequencing, whole-genome shotgun sequencing, and Hi-C sequencing, respectively.

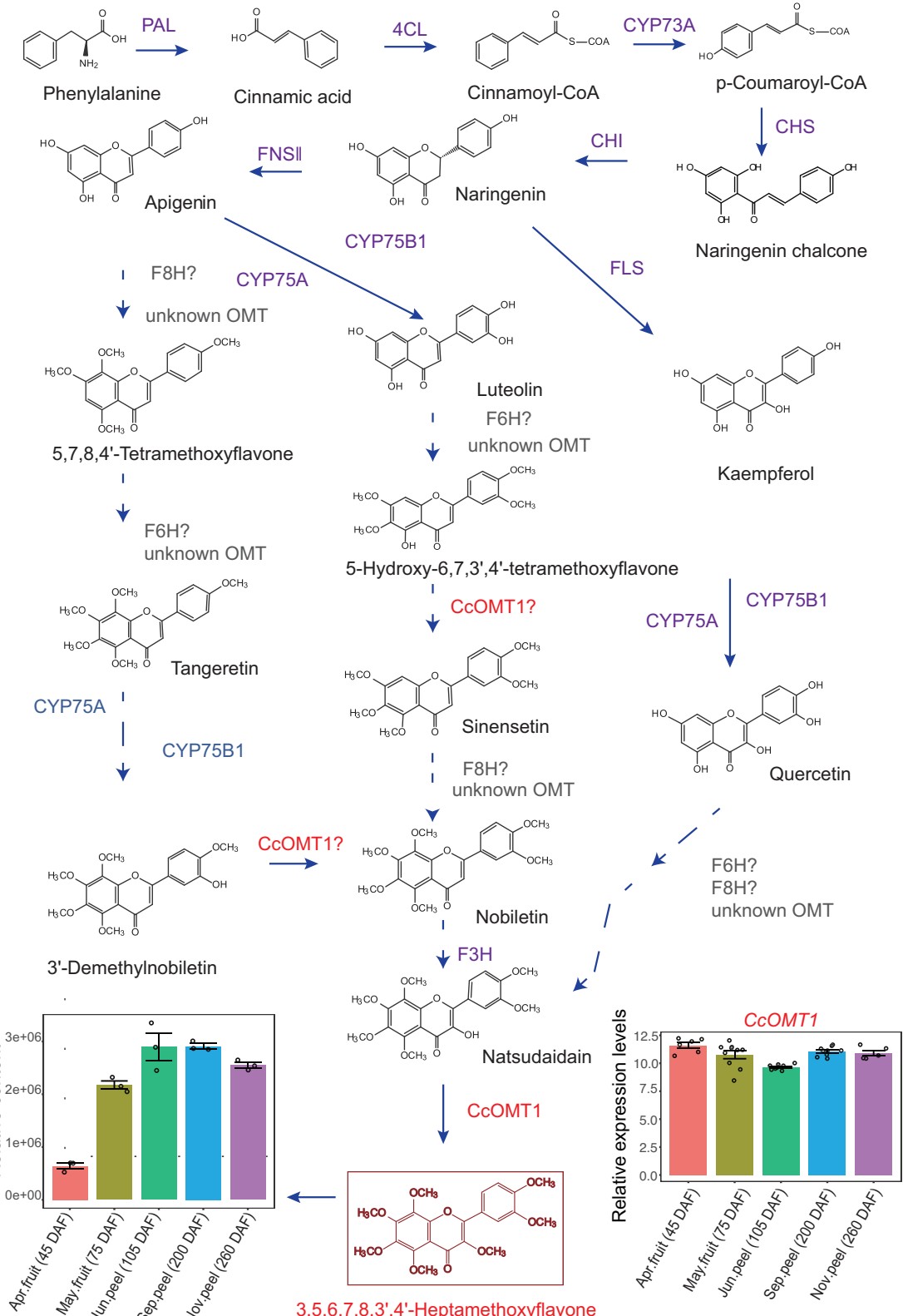

**Fig. 8 | Proposed biosynthetic pathway of PMF in CRC.** The dashed arrow indicates an uncertain pathway. CHS chalcone synthase, CHI chalcone isomerase, CYP75A flavonoid 3',5'-hydroxylase, CYP75B1 flavonoid 3'-monooxygenase, FNSII flavone synthase, F3H flavanone-3-hydroxylase, F6H flavanone-6-hydroxylase, F8H flavanone-8-hydroxylase, FLS flavonol synthase. Error bars, mean ± s.d. For RNA-seq samples of Apr fruit, May fruit, Jun peel, Sep peel, and Nov peel, n = 6,10,9,9, and 4, respectively. For metabolomics samples, n = 3.

A shotgun library with an insert size of 500 bp was prepared from genomic DNA using the BGISEQ Sequencing Kit following the manufacturer's standard protocol, and 52 Gb of sequencing data was produced by BGISEQ platform with PE length of 100 bp.

For the nanopore sequencing library construction, the genomic DNA was sheared, and longer fragments were selected using AMPure XP beads. After purification, the DNA fragments were repaired for ligation of barcodes/adapters. Finally, the library was sequenced and approximately 40 GB of filtered long–reads data was generated from the PromethION Nanopore sequencer.

For Hi-C library construction, fresh young leaves of CRC in February were cross-linked with formaldehyde, and then DNA was extracted by the CTAB method[63] after nucleus extraction, enzymatic digestion, end biotin labeling, DNA ligase ligation, protease digestion of ligation sites, and end debiotinization. DNA fragmentation was performed by using a Covaris LE220 interrupter to recover fragments between 150-300 bp and purify these fragments. The library was then constructed by DNA end repair, adenine end addition, junctions ligation and PCR amplification and purification, and then subjected to BGISEQ platform sequencing with PE 100 strategy. A total of 35 GB Hi-C data was produced.

## Genome assembly

A total of 40 Gb cleaned short read data was used to estimate the genome size of CRC by GenomeScope (v1.0) with 'k = 17'[64]. A total of 40 Gb long read data from Nanopore sequencing were used to assemble the primary assembly using Nextdenovo (v2.2 beta.0) (https://github.com/Nextomics/NextDenovo) with the parameter "task=all (correct and assemble)".

In the reads self-correction step of Nextdenovo, the read_cutoff was set to 1k, and the seed_cutoff was set to 36,685 bp. Next, the long and the short clean reads were used to correct and polish the assembly with Nextpolish (v1.3.0)[65] with the general parameter "task = best". After that, the genome was assembled into scaffolds with a total length of 314.96 Mb. Purge_dups (v1.2.5)[66] was used to assess the duplication of the genome assembly.

The 3D-DNA (v180922)[67] pipeline was used to construct the chromosome-level assembly with the Hi-C data. To obtain the final assembly, the visualization tool Juicebox was used to improve the intermediate assembly by hand. The quality of the genome assembly was evaluated by BUSCO (v3.01, embryophyta_odb10, 1375)[68]. The cleaned short reads were aligned to the genome assembly to assess the genome accuracy via BWA[69].

## Genome annotation

An integrated strategy that combines de novo and homology-based methods was used to annotate repetitive sequences. PILER (v1.0) (http://www.drive5.com/piler), RepeatScout (v1.0.5) (http://www.repeatmasker.org) and LTR_FINDER (v1.06)[70] were used to produce de novo predicted results. RepeatClassifier (version 1.0.8; https://repeatmasker.org/RepeatModeler/) was also used to classify the results. Next, these results were merged and used to construct a library as an input file for RepeatMasker (open-4.0.6) (http://www.repeatmasker.org). We also used Tandem Repeats Finder (v4.07b)[71] to detect tandem repeats. For homology-based prediction, the genome sequences were searched against Repbase database (https://www.girinst.org/repbase/) by RepeatProteinMask (http://www.repeatmasker.org) and RepeatMasker.

We combined three approaches, including de novo based, homology-based prediction and RNA-seq-based methods to predict gene models via BRAKER2 (v2.1.5)[72]. We constructed a homology-based protein database including protein sequences from *C. sinensis*, *C. celemintina*, *C. reticulata*, *C. maxima* and *C. unshiu*. The protein database, as well as RNA-seq data of leaves and peels from CRC, were used as extrinsic evidence data for GeneMark-ETP and AUGUSTUS training in the BRAKER2 pipeline[73]. The RNA-seq data were aligned against the genome by TopHat2 (version 2.1.0)[74] before running the BRAKER2 pipeline. To evaluate the quality of the gene set, BUSCO[68] was used to check completeness.

For functional annotation, all the predicted genes were aligned against six databases, including KEGG[75], NR (https://www.ncbi.nlm.nih.gov/), SwissProt (https://www.uniprot.org/), Trembl (https://www.uniprot.org/), InterPro (https://www.ebi.ac.uk/interpro/), and COG[76].

The program iTak (v1.4)[77] was used to detect transcription factors in the CRC genome.

The annotation of tRNAs was performed by tRNAscan-SE (v1.3.1)[78]. rRNA sequences from *A.thaliana* and *O.sativa* were used as reference sequences to annotate rRNA. The database used for miRNA and snRNA annotation was Rfam (v12.0)[79]. BLAST (v2.2.26) was used for the mapping of miRNAs, snRNAs and rRNAs with the parameters "-p blastn -W 7 -e 1 -v 10000 -b 10000 -m8".

## Phylogenetic analysis

To construct a phylogenetic tree of CRC and other species, 9 citrus and citrus-related species, namely, *A. buxifolia*, *C. clementina*, *C. ichangensis*, *C. maxima*, *C. medica*, *C. reticulata*, *C. sinensis*, *C. unshiu* and *P. trifoliata* were selected, and *A. trichopoda*, *O. sativa*, *V. vinifera*, *A. thaliana*, and *T. cacao* were used as outgroup species. The protein sequences of these species were downloaded from public databases. OrthoFinder (v2.5.2)[80] was used to infer orthogroups among these 15 species. A maximum-likelihood phylogenetic tree was constructed based on these single-copy orthologs using MAFFT (v.7.310)[81] and RAxML (v8.2.12)[82] with 1000 bootstraps and the model "-m PROTGAMMAJTTF".

To estimate the divergence times of the 15 species, the MCMC tree program (http://abacus.gene.ucl.ac.uk/software/paml.html) implemented in Phylogenetic Analysis by Maximum Likelihood (PAML) was used.

The analysis of the expansion and contraction of the gene families was performed on CAFE5 (v1.1)[83].

## Synteny and whole-genome duplication analysis

The MCscan pipeline[84] was used to infer intergenomic synteny among CRC, *C. sinensis* and *C. clementina*. To detect ancient whole-genome duplication events, wgd (v1.1.0)[85] was used. To obtain more accurate results, DupGen_finder[86] was used to identify gene pairs derived from WGD events. The number of synonymous substitutions per synonymous site (Ks) between these gene pairs were calculated. In addition, the four-fold synonymous third-codon transversion rate (4DTv) was also calculated in CRC and *C. sinensis*.

## Identifying candidate functional genes related to flavonoid synthesis

All candidate genes were searched in three steps. First, known sequences collected from public databases served as queries in the BLASTP search (<1e−05). Second, related HMMER domain structure models were downloaded and were used for the HMMER search. Third, all the results from the above steps were confirmed by CDD (https://www.ncbi.nlm.nih.gov/Structure/bwrpsb/bwrpsb.cgi) and SMART. Only sequences with the corresponding complete domains were considered candidate genes.

To search for candidate OMT genes in the CRC genome, we collected OMT genes from other *citrus* species, such as *C. depressa* and *C. unshiu*, and reviewed OMT entries from UniProt, including *A. thaliana*, *O. sativa*, and *N. tabacum*. For the HMMER search, the *O*-methyltransferase domain models PF00891 and PF01596 were downloaded from the Pfam database (http://pfam-legacy.xfam.org/).

A similar method was applied to identify UGT genes. We collected UGT sequences of *A. thaliana*, *C. sinensis*, *C. maxima*, *O. sativa*, *Z. mays*, *P. granatum*, *G. biloba* and *C. roseus* from UniProt (https://www.uniprot.org/) and NCBI.

## Transcriptome sequencing

For transcriptome sequencing, organs and tissue samples (leaves, flowers, young fruits, peels, pulp, seeds) from different developmental stages of CRC were collected from the same fruit tree monthly in Xinhui. Young flowers were collected in February and young fruits were collected at 45 DAF and 75 DAF in 2021. The mandarin peels, pulp, seeds, and leaves were collected from 105 DAF to 260 DAF in the same year. At least three biological replicates of each sample were collected.

Total RNA was extracted from all these organs and tissues using Column Plant RNAout Kit according to the instructions (TIANDZ, China). After integrity assessment using NanoDrop (Thermo, USA) and 2100 Agilent Bioanalyzer (Agilent, USA), 203 RNA samples were qualified for library construction. A total of 1 ug qualified RNA from each sample was used to construct the BGI-based mRNA-seq library. In general, mRNA molecules were purified using oligo(dT)-attached magnetic beads and then fragmented into small pieces. The cDNA was generated using random hexamer-primed reverse transcription and adaptors were ligated to the ends of these 3' adenylated cDNA fragments. Then, the library was sequenced on the BGISEQ platform using a 100 bp pair-ended strategy.

The cleaned reads were obtained after removing adapters and filtering low-quality sequences by Trimmomatic (v0.38)[87]. Reads were aligned to the genome using STAR (v2.7.9a) in 2-pass mode[88]. After read alignment, the mapped reads counted by featureCounts[89]. The R package DESeq2[90] was used for raw count data normalization and gene differential expression analysis. Gene Ontology (GO) and Kyoto Encyclopedia of Genes and Genomes (KEGG) pathway enrichment analyses of the DEGs were using clusterProfiler 4.0[91].

The transcriptome data of citrus peels from the core region and other regions were produced in our previous study[92]. Three representative orchards (Tianma, Wubao and Maichong) in Xinhui (geo-authentic product region) of Guangdong Province were selected as the core region. Three orchards Taishan from Jiangmen city, Longmen from Huizhou city of Guangdong Province, Daan from Guangxi Province were selected to represent the noncore region.

## Co-expression analysis

The WGCNA package (v1.70-3)[93] was used to conduct weighted correlation network analysis among 15 transcriptome samples from 5 different developmental stages, namely, 45 DAF-fruit, 75 DAF-fruit, 105 DAF-peel, 200 DAF-peel and 260 DAF-peel, with three biological replicates for each developmental stage. Cytoscape (v3.9.0)[94] was used for the analysis of gene co-expression network and visualization.

## Metabolite detection and data analysis

The above 15 samples including fruits and peels were frozen and dried in a freezer separately before being ground into powder by a grinder for 90 s at 30 Hz. Then, 100 mg powder of each sample was dissolved in 1.2 mL of 70% methanol extract. The mixture was vortexed for 30 s and placed 30 min. This operation was repeated 6 times. Then, the mixture of each sample was placed in a refrigerator at 4 °C overnight. Next, each mixture was centrifuged for 10 min at $11{,}304 \times g$ and the supernatant was filtered through a 0.22 μm pore size microporous membrane. The filtrate was transferred into a detection bottle and used for UPLC-MS/MS analysis. The analyzer system included Ultra Performance Liquid Chromatography (UPLC) (SHIMADZU Nexera X2, Kyoto, Japan) and Tandem Mass Spectrometry (MS/MS) (Applied Biosystems 4500 QTRAP, AB Sciex, Framingham, MA, USA). Qualitative analysis of the compounds was performed based on a self-construction database and quantitative analysis relied on multiple reaction monitoring (MRM) mode. Analyst 1.6.3 (https://sciex.com/products/software/analystsoftware) was used for processing the mass spectrometry data.

Orthogonal Projections to Latent Structures Discriminant Analysis (OPLS-DA) was used to discriminate the differences among the sample groups. Differentially accumulated metabolites were identified using Variable Importance in Projection (VIP) with the following thresholds: VIP ≥ 1 and a fold change ≥ 2 or fold change ≤ 0.5.

## Correlation analysis between DEGs and DMs

The R package Hmisc (v4.60) (https://hbiostat.org/R/Hmisc/) was used to calculate the Pearson correlation coefficient between DEGs and DMs. The correlations of the DEGs and DMs with a coefficient > 0.8 and a false discovery rate (FDR) < 0.05 would be used for further analysis. The correlation network among the DEGs and DMs was constructed using Cytoscape (v3.9.0)[94].

## Protein expression and purification

The full length of *CcOMT1* cDNA was amplified by PCR using a primer pair (forward primer: CCGGATCCATGGGTTCAACCAGTTCAGAAA; reverse primer: CCCTCGAGTCAAGCACTCTTGAGAAATTCCATA) at 95 °C for 3 min, 40 cycles of 95 °C for 30 s, and 58 °C for 30 s, and then 68 °C for 1.5 min, followed by 68 °C for 8 min. The amplified fragment was then inserted into the pET-28a (+) vector. The gene was sequenced by GenScript Ltd (Nanjing, China). The recombinant plasmid pET28a (+)-CcOMT1 was transformed into *E. coli* BL21(DE3) (Weidi Biotech, China) for heterologous expression. The *E. coli* cells were grown in 300 mL LB medium with kanamycin (50 μg/mL) at 37 °C and 200 rpm. After $OD_{600}$ reached 0.6, the cells were induced with 0.5 mM isopropyl β-D-thiogalactoside (IPTG) at 16 °C and 160 rpm for another 20 h. The cells were harvested by centrifugation at $4000 \times g$ for 10 min at 4 °C, and then resuspended in 10 mL binding buffer (50 mM $NaH_2PO_4$, 300 mM NaCl, 10 mM imidazole, pH 8.0) containing 1 mM phenylmethylsulfonyl fluoride (PMSF). Then, cells were disrupted by sonication in an ice bath, and the cell debris was removed by centrifugation at $4000 \times g$ for 45 min at 4 °C. The supernatant was collected and sampled to 0.5 mL Ni-NTA agarose (Qiagen, Germany), which was loaded in a column that was prebalanced with a binding buffer at 4 °C for 1.5 h. The resin was then eluted with wash buffer (50 mM $NaH_2PO_4$, 300 mM NaCl, 20 mM imidazole, pH 8.0). Elution was performed with different concentrations of elution buffer (50 mM NaH2PO4, 300 mM NaCl, 250 mM imidazole, pH 8.0). Furthermore, to obtain higher purity enzymes, Ni-NTA protein was purified using a centrifugal concentrator with Amicon Ultra-30K (Millipore) for further purification. According to SDS-PAGE analysis, the purity of the target protein was > 95%, and protein concentration was determined by the Protein Quantitative Kit (TransGen Biotech, China) using bovine serum albumin (BSA) as a standard. Finally, 20% glycerol (pre-cooled) was added to the purified protein, mixed well, and stored at -80 °C.

## In vitro assays of CcOMT1

The function of CcOMT1 was characterized by co-incubating 0.3 mg purified protein, 0.2 mM natsudaidain, 5 mM S-adenosyl-l-methionine (SAM), 1 mM DTT, and 1% glycerol (v/v) in 300 μL of 25 mM $NaH_2PO_4$-$Na_2HPO_4$ buffer (pH 6.0). Incubated the reaction mixture for 4 h at 50 °C with shaking and then terminated by adding 200 μL of methanol. After the product was evaporated to dryness, the residue was dissolved in ethanol and 20% DMSO. The mixture was centrifuged at $16{,}200 \times g$ for 10 min. The supernatants were analyzed by HPLC and LC-QTOF−MS. The HPLC analysis was performed on a BDS HYPERSIL C18 column (4.6 mm × 250 mm, 5 μm) at a flow rate of 1 mL/min. The mobile phase consisted of (A) 0.1% formic-water and (B) 100% acetonitrile. The gradient elution consisted of 4 stages: (1) 0−3 min, 25−50% B; (2) 3−7 min, 50−58% B; (3) 7−11 min, 58−58% B, (4)11−15 min, 58−70% B. The column temperature was set at 30 °C and the wavelength was 330 nm[13].

The biochemical properties of CcOMT1 were studied using natsudaidain as a methyl receptor and SAM as a methyl donor. To study the optimal reaction temperature for CcOMT1 activity, the reactions were incubated at different temperatures (16, 25, 37, 42, 50, 55, 60, 65, and 70 °C). To determine the optimal pH values for CcOMT1 activity,

the pH values were 4.0–6.0 (citric acid-sodium citrate buffer), 6.0–8.0 ($Na_2HPO_4$-$NaH_2PO_4$ buffer), 7.0–9.0 (Tris-HCl buffer) and 9.0–11.0 ($Na_2CO_3$-$NaHCO_3$ buffer) in different reaction buffers. For each of these conditions, the experiments were repeated in triplicate. Reactions were stopped by adding 200 µL of methanol. After the product was evaporated to dryness, the residue was dissolved in ethanol and 20% DMSO. The mixture was centrifuged at $16,200 \times g$ for 10 min. Supernatants were detected by HPLC as described above.

### Kinetics measurement

For kinetic studies of CcOMT1 and its variant CcOMT1-W141Y toward natsudaidain, an enzymatic assay containing 25 mM $NaH_2PO_4$-$Na_2HPO4$ buffer (pH 6.0), 0.26 mg of purified CcOMT1, 5 mM S-adenosyl-l-methionine (SAM), 1 mM DTT, 1% glycerol (v/v) and varying concentrations (25 to 200 µM or 5 to 100 µM) of natsudaidain, was performed at 50 °C for 2 h in a final volume of 300 µL. All the reactions were terminated by methanol and centrifuged at $16200 \times g$ for 10 min. Supernatants were analyzed by HPLC as described above. All experiments were performed in triplicate. The Michaelis–Menten constant (Km) was determined from non-linear regression fitting, and the turnover number (Kcat) was calculated.

### Substrate promiscuity of CcOMT1

All reactions were individually conducted in a final volume of 300 µL containing 25 mM $NaH_2PO_4$-$Na_2HPO_4$ buffer (pH 6.0), 0.2 mM flavonoid substrates, 5 mM methyl donor, 1 mM DTT, 1% glycerol (v/v) and 0.3 mg of purified CcOMT1. The reactions were incubated at 50 °C for around 4 h and were terminated by adding 200 µL methanol. After centrifugation at $16,200 \times g$ for 10 min, the supernatants were analyzed by HPLC as described above.

### Molecular docking and mutagenesis

CcOMT1 was modeled using Robbeta (https://robetta.bakerlab.org/submit.php). The active pocket of the protein was predicted by Pocasa (https://g6altair.sci.hokudai.ac.jp/g6/service/pocasa/). Molecule docking of CcOMT1 with natsudaidain performed by Autodock Vina version 1.5.6[95]. After docking, the conformation with the lowest binding energy was selected for further study. The model was then viewed and rendered using PyMOL. The residues within 5 Å of any ligand position were refined. Site-directed mutagenesis was performed on CcOMT1 using QuickMutation™ Site-Directed Mutagenesis Kit (Beyotime,China) according to the manufacturer's instructions. The mutation sites included W141A, Y142A, F164A, N174A, S182A, W141F, W141Y, Y142F, Y142S, F164S, F164L, N174V, N174S, S182L, S182G and S182W. Primers for site-directed mutagenesis are listed in Supplementary Data 14. The enzyme activity analysis of the mutant recombinant proteins were the same as that of CcOMT1.

### Transient expression and of CcOMT1 in CRC in vivo

The target gene *CcOMT1* was amplified using the primer set of CcOMT1-pCAMBIA1301 (Supplementary Data 15) and inserted into the expression vector PCAMBIA1301. *A. tumefaciens* strains carrying the PCAMBIA1301 plasmid vector with *CcOMT1* and GFP genes were grown on LB medium for more than 48 h and then resuspended in infiltration buffer (10 mM $MgCl_2$, 10 mM MES, 150 mM acetosyringone; pH 5.6) at an OD600 of 0.8. The fruits of CRC used for infiltration were harvested from an orchard in Xinhui, Guangdong, China on October 20th of 2022 (230 DAF). The suspensions of *A. tumefaciens* strains expressing the target gene *CcOMT1* and the empty PCAMBIA1301 plasmid vector (as a control) were injected into the peel on opposite sides of the equatorial plane of the same fruit[96]. After infiltration, the fruits were kept in the dark for 1 day and maintained under 16 h light/8 h dark conditions and then the peels of the injection region were sampled to analyze the metabolites and gene expression after 5 days by HPLC and RT-qPCR, respectively.

Total RNA was extracted from the tested citus fruits using Fas-tPure Plant Total RNA Isolation Kit (Vazyme, Nanjing, China). HiScript®III 1st Strand cDNA Synthesis Kit (+gDNA wiper) (vazyme, Nanjing, China) was used to remove genomic DNA contamination from total RNA and to synthesize cDNA. Then RT-qPCR was conducted to detect the gene expression of *CcOMT1* according to the manufacturer's protocol. The primers used for RT-qPCR are listed in Supplementary Data 15.

The flavonoids in the tested citrus peels were extracted by ultrasound-asisited extraction, and the peels were ground into powder in liquid nitrogen. After extraction with 2 ml methanol solution, the peels were ultrasounded at 40°C for 40 min, and centrifuged at $12,000 \times g$ for 30 min. The supernatant was filtered through 0.22 uM organic filter membrane[25]. An HPLC-DAD system (G1316A quaternary pump, G1315C diode array detector; Waters Crop., Milford MA) coupled with an Agilent C18 ODS column (4.6×250 mm, 5 µm) was used for flavonoid detection. The compounds were separated at room temperature using the following gradients of acetonitrile (solvent A) versus 0.2% (v/v) formic acid in water (solvent B) at a flow rate of 1 mL/min: 0–10 min, 25%–40% A; 10–15 min, 40%–60% A; 15–18 min, 60%–80% A; 18–22 min, 80%–25% A; 22–25 min, 25% A[25]. Flavonoids were monitored at 340 nm, and the concentration of flavonoids was calculated by using a standard curve method.

### TRV-mediated virus-induced gene silencing in citrus fruits

TRV-mediated virus-induced gene silencing (VIGS) was performed for transient silencing of targeted genes[97]. For the construction of TRV2-CcOMT1, a fragment of the *CcOMT1* ORF (300, 551-851 bp) was amplified by PCR with the TRV2-CcOMT1-FP and TRV2-CcOMT1-RP primer pair (Supplementary Data 15), which was then introduced into the TRV2 vector. The empty vector was used as a control. All the vectors were introduced into *A. tumefaciens* strain GV3101.The *Agrobacterium* cells harboring the TRV1 and TRV2 recombinants were grown in LB broth to OD600 = 1.0[27], and then *Agrobacterium* cultures were centrifuged at $6000 \times g$ for 10 min and resuspended in infiltration buffer (10 mM $MgCl_2$, 10 mM MES, 200 µM acetosyringone, 0.01% Tween20; pH 5.6). Resuspended *A. tumefaciens* TRV1 and TRV2-*CcOMT1* were mixed at a 1:1 ratio and infiltrated into the citrus fruit on the trees, three single-tree-based biological replicates were set up[98]. Citrus fruits were collected at two weeks after infiltration. Each treatment had three biological replicates with a minimum of 20 citrus fruits. These samples were analyzed for flavonoid content and gene expression as described above.

### Reporting summary

Further information on research design is available in the Nature Portfolio Reporting Summary linked to this article.

## Data availability

The raw RNA-seq data and genome sequences have been deposited in the Genome Sequence Archive (GSA) in National Genomics Data Center, China National Center for Bioinformation/Beijing Institute of Genomics, Chinese Academy of Sciences under accession number CRA015571. The chromosome-level genome assembly and the scaffold-level genome assembly data have been deposited in the Genome Warehouse under accession numbers GWHERQK00000000 [https://ngdc.cncb.ac.cn/gwh/Assembly/84113/show] and GWHERPZ00000000 [https://ngdc.cncb.ac.cn/gwh/Assembly/84102/show], respectively. The transcriptome, genome sequence and genome assembly data have also been deposited in CNGBdb under accession codes CNP0003922 and CNP0003929, respectively. The metabolomics data have been deposited to the EMBL-EBI MetaboLights database with the identifier MTBLS9832. Source data are provided with this paper.

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

## Acknowledgements

The research was supported by the Key Realm R&D Program of Guangdong Province (2020B020221001 to HW), and the Open Competition Program of Ten Major Directions of Agricultural Science and Technology Innovation for the 14th Five-Year Plan of Guangdong Province (2022SDZG07 to HW), and the Major Science and Technology Projects of Yunnan Province (No. 860 202002AA100007, H.L.), and the Laboratory of Lingnan Modern Agriculture Project (NZ 2021024 to H.W.), and the Key Laboratory of Genomics, Ministry of Agriculture, BGI-Shenzhen, Shenzhen 518120, China; Guangdong Provincial Key Laboratory of Core Collection of Crop Genetic Resources Research and Application, BGI-Shenzhen, Shenzhen 518120, China; and the Shenzhen Engineering laboratory of Crop Molecular Design Breeding, BGI-Shenzhen, Shenzhen 518120, China.

## Author contributions

H.L., E.L., H.W., Jiawen Wen and Y.W. established the concept of the study. Jiawen Wen, Y.W., Xu Lu, H.P., D.J., Jialing Wen, C.J., J.S., Xinyue Luo, X.J. and J.Z. collected and processed the samples. Jiawen Wen performed the bioinformatics analysis. Xu Lu and D.J. performed protein expression and purification, in vitro assays, molecular docking and mutagenesis experiments, and kinetics measurement. H.P. performed the transient overexpression and virus-induced gene silencing experiments. Jiawen Wen, Y.W., Xu Lu, D.J. and H.P. wrote the draft. H. L., E.L., H.W. and S. K. S. revised and edited the manuscript. All authors have discussed the results, read, and approved the contents of the manuscript.

## Competing interests

The authors declare no competing interests.
