## [Peer Review File · Nature Communications]

An integrated multi-omics approach reveals polymethoxylated flavonoid biosynthesis in *Citrus reticulata* cv. ChachiensisReviewer #1 (Remarks to the Author):

The manuscript by Wen et al assembled a high quality genome of *Citrus Chachiensis*, a landrace for medicinal purpose. Transcriptome and metabolome analysis provide insight into the biosynthesis of PMFs in the fruits of *Chachiensis*. Meanwhile, the authors also identified key candidate genes for the PMF accumulation. Generally, this study provides useful data for the study of citrus community, and new information on the regulation of PMFs. I offer the below revisions for the improvements of this ms.

Major concern

1. The key point of OMT need to be consolidated to support the conclusion of this study. One weak issue is the function of OMT1, which was proposed based on transient assay, it needs to be confirmed by gene editing or RNAi, or stable overexpression experiments.
2. Line 228, CZG-jg206 was proposed to be the main gene for the formation of bitter taste in *C. chachiensis*. Actually, this gene may not functional in relative to Cm1,2RhaT. The authors may need to confirm neohesperidoside by LC-MS and standards in *C. chachiensis*.
3. UGT seems distracted the logic of this study. UGTs are associated with catalyzation of many metabolites. So the identified 96 UGTs are a pool of many kinds of function. Specifically, function of UGTs or specific UGTs for the PMFs of *C. chachiensis* need to be confirmed.
4. The details for the metabolic data such as secondary mass spectrometry for substance qualitative identification, molecular ion fragment information. The instrument used for different purpose are different.

It seems the language of this manuscript need to be improved.

Minor points

5. The sampling information of *C. chachiensis* from different places need to be clarified.
6. Line 54 to 62, it is not necessary to introduce the regulation of anthocyanin, the key point of this article is PMF.
7. Line 206-207, 639-642, the sampling time were described as August, September, October, November, should be corrected as day after flowering (DAF)
8. Line 10 and 89, please indicate whether Contig N50 or Scaffold N50? Contig N50 could represent genome quality better than Scaffold N50.
9. Line 142, Fig 2a can not reflect your result "*C. chachiensis* is related more closely to *C. clementina* than to *C. reticulata* (mangshan wild mandarin)", maybe you intend to cite Fig 2c, please check.
10. *C. ichangensis* was one of the ancestors of *Citrus*, its position in Fig 2c clustering with *C. maxima* and *C. medica* is unusual. This node actually is somewhat sensitive to the data.
11. Line 156 and 158, the duplication event shared by all eudicot plants were WGT- γ , and Fig 2e did reflect the duplication event, but the red circles you marked were not apparent to demonstrate it, please refer to the Fig 2 of sunflower genome article (Badouin et al., 2017).
12. Line 153, it is an older WGD event, maybe you mean "ancient WGD event".
13. Line 187 to 188, the reference of gene CICLE_v10026344mg should be added.
14. Line 216 PMF glycosylation? It was believed that this step was not happen in *C. chachiensis*.
15. Line 309 accumulated the highest PMF content followed by June, may need to confirm.
16. Figure S10, the unit for heptamethoxyflavone is irregular.
17. Table S1 should have units.
18. line 23 in the abstract, *C. clemintina* is not correct.

Reviewer #2 (Remarks to the Author):

In this manuscript, the authors reported chromosome-level reference genome of *Citrus chachiensis*, the transcriptomic and metabolomic analysis towards exploring the regulatory networks and catalytic enzymes responsible for the biosynthesis of polymethylated flavonoids in *Citrus*, as well as the characterization of a O-methyltransferase that exhibits the detectable activity of 3-O-mthyalton of PMF in vitro. The manuscript was compiled with quite lots of data; however, unfortunately, the manuscript was poorly developed, and consequently its overall quality and readability is low. Many editorial and scientific issues pertaining to the text description, the experimental designs, and the data presentation can be found throughout the manuscript. For instance, the manuscript has cumbersome or redundant description, lacks structural integrity,

contains scientific errors/inaccuracy, and has poor data presentation.

I mention a few of such issues, more can be found throughout the manuscript.

1) In the Introduction, it missed the information on the current research advance of Citrus genome sequencing, i.e., how many citrus species/varieties have been reported for their genomes and then the authors should link to their rationale conducting genome sequencing for *C. chachiensis*. In manuscript, the authors claimed that exploring *C. chachiensis* genome aimed to elucidate PMF biosynthesis and regulation. However, the authors also stated that PMFs are most abundantly enriched in the peel of *C. sinensis* and *C. reticulata*. Then it is not clear why the authors had to explore the genome of *C. chachiensis* but not those of *C. sinensis* and *C. reticulata*?

2) The following description (Lin 86-89) is scientifically wrong: "one of which is caffeoyl-CoA O-methyltransferase (COMT) with a wide range of substrates, including myoinositol, chalcones, caffeic acid, and so on. The other group is caffeoyl-CoA O-methyltransferase (CCoAOMT), which needs ion catalyzes in the reaction". COMT represents caffeic acid 3/5-O-methyltransferase. Also, please check the grammar for the last sentence.

3) In the following descriptions: (Line 348-352) "AP2-ERF/ERF transcription factors were hub genes and strongly associated with the flavonoid biosynthesis genes (HCT: hydroxycinnamoyltransferase, FLS) and UDP-glycosyltransferase genes (UGTs), indicating that AP2-ERF/ERF transcription factors might modulate the flavonoid synthesis by regulating the HCT, FLS and UGT genes"; (Line 363-365) "AP2/ERF-ERF, UGT as well as HCT were identified as the key gene in the regulatory network, which might drive the related gene expression and promoting the accumulation of PMFs"; (Line 441) "Early flavonoid biosynthesis genes such as HCT, PAL (phenylalanine ammonia-lyase), CHS, and CHI contributed to naringenin production...", the authors claimed HCT is a key gene for flavonoid synthesis. This is not correct. HCT per se encodes an enzyme involved in monolignol synthesis and has nothing with flavonoid formation. It might be possible that some aromatic acyltransferases involved in the flavonoid glucoside modification but definitely they are not the ones involved in the early flavonoid biosynthetic steps.

4) Regarding metabolomic analysis, the authors described in the Results that the metabolite extracts were analyzed by GC-MS; however, in the Method the LC-MS methods were described.

5) The authors reported that metabolomic study revealed 94 flavones, 75 flavonols, 49 flavonoid carbonoside, 49 flavanones, 14 tannins, 12 chalcones, 4 dihydroflavonols and others. It is better also present the detail metabolites list and their chemical assignments in a supplemental datasheet or the figures, and be cited within the main text. In addition, the method on chemical identification/assignments of these detected flavonoids should be provided in the Method section.

6) The exploration of CcOMT1 in the context surprisingly disconnected from the described transcriptomic and metabolic studies. The authors stated that CcOMT1 was identified from protein database by sequence similarity searching with a tomato flavonoid 3-OMT, ShMOMT3, which appears that the CcOMT1 characterization was conducted parallelly with the described omics studies, and those different studies were just simply compiled together, but lack logical connection and integration.

7) Regarding the characterization of CcOMT1, a few expected experiments were missing: To determine a catalytic enzyme and compare it with the others (or its mutant variants), the kinetic parameters should be measured under the apparent optimal conditions; moreover, since phenolic OMTs often exhibit promiscuity to their substrates and with regio-specificity, the enzymatic assays of CcOMT1 should be conducted with a broader phenolics, including those bearing the hydroxy moiety at different positions.

8) The CcOMT1 expression pattern should be highlighted in a supplemental figure.

9) Line 30-31, "CcOMT1 could catalyze natsudaoidain to produce 3,5,6,7,8,3',4'-heptamethoxyflavone by adding a methyl donor SAM". What does it mean "adding a methyl donor

SAM”?

10) Line 746-747, “The biochemical properties of CcOMT1 were studied using natsudaidain as a sugar receptor and SAM as a sugar donor”---what do they mean “sugar receptor” and “sugar donor”?? CcOMT is a methyltransferase not a glucosyltransferase, and SAM is a methyl donor.

11) The sentence needs to be rephrased “the limits of knowledge is to know the transcriptional regulation network and the biosynthetic pathway of PMF”

12) “illuminate the key OMTs”----might be better to change it as “elucidate the key OMTs”

13) The sentence (line 110—112) “The estimated genome size of *C. chachiensis* was predicted to be 284.29 Mb with 1.01% heterozygosity ratio based on the analysis of k-mer depth distribution by short-read data (Figure 1a; Figure S1).” was pointed to the Figure 1a. However, the figure 1a shows nothing with *C. chachiensis* genome.

14) Line 246-247, “These CGT genes exhibited higher expression levels in fruits and peel, which would contribute to higher content of corresponding products in fruits and peel”. The CGT gene expression data should be presented to support the statement.

15) The description similar to the line 384-386 “3,5,6,7,8,3',4'-heptamethoxyflavone is one of the most important PMFs with multiple medicinal effects...” is repeated several times in the text.

16) Line 343—347, “The MEyellow module was highly correlated with near mature peel in September ($r=0.89$, $P=9e-06$) and significantly positive correlated with PMFs ($r=0.7$, $P=0.004$) and flavonoid C-glycoside ($r=0.8$, $P=3e-04$). The MEblue module was correlated with young fruits in April and negatively correlated with flavonols ($r=-0.7$, $P=0.004$) and flavonoid-O-glucosides ($r=-0.67$, $P=0.006$)”, this description should be accompanied with a figure presentation.

17) The uploaded main figures have so low resolution and their contents are barely observable.

Dear Reviewers,

We greatly appreciate your constructive comments and suggestions. We have performed additional experiments and analysis that have further validated and strengthened all of our major conclusions in the manuscript. Below we provide comments on each of their suggestions or concerns.

Reviewer #1 (Remarks to the Author):

The manuscript by Wen et al assembled a high quality genome of Citrus Chachiensis, a landrace for medicinal purpose. Transcriptome and metabolome analysis provide insight into the biosynthesis of PMFs in the fruits of Chachiensis. Meanwhile, the authors also identified key candidate genes for the PMF accumulation. Generally, this study provides useful data for the study of citrus community, and new information on the regulation of PMFs. I offer the below revisions for the improvements of this ms.

Major concern

1. The key point of OMT need to be consolidated to support the conclusion of this study. One weak issue is the function of OMT1, which was proposed based on transient assay, it needs to be confirmed by gene editing or RNAi, or stable overexpression experiments.

Response: Thanks for your constructive comment. The virus-induced gene silencing (VIGS) system was used to confirm the functions of *CcOMT1* according to your suggestion. Silencing of *CcOMT1* significantly reduced the expression level of *CcOMT1* and the accumulation level of 3,5,6,7,8,3',4'-heptamethoxyflavone in *C. chachiensis* fruits. The results of transient overexpression and silencing of *CcOMT1* further demonstrated that *CcOMT1* was involved in the biosynthesis of 3,5,6,7,8,3',4'-heptamethoxyflavone (Figure 6).

I updated the results in the line 363-375 as follow:

“For further confirmation of the potential role of CcOMT1 in the methylation of flavonoids *in vivo*, transient overexpression experiments were conducted. The transcript levels of *CcOMT1* and the content of 3,5,6,7,8,3',4'-heptamethoxyflavone in *C. chachiensis* peel infiltrated with *CcOMT1*-pCAMBIA1301 were both increased significantly compared to the empty vector (Figure 6a and 6b). The virus-induced gene silencing (VIGS) system was also used to further confirm the functions of *CcOMT1*. *Agrobacterium tumefaciens* cultures with TRV1/TRV2 or TRV1/TRV2-*CcOMT1* constructs were transiently infiltrated into *C. chachiensis* fruits. Compared with the empty vector (TRV1/TRV2), VIGS significantly reduced the expression level of *CcOMT1* and the accumulation level of 3,5,6,7,8,3',4'-heptamethoxyflavone (Figure 6c and 6d). The results of transient overexpression and silencing of *CcOMT1* demonstrated that CcOMT1 was involved in the biosynthesis of 3,5,6,7,8,3',4'-heptamethoxyflavone in *C. chachiensis*.”

The experiment method in line 884 and 896 as follows:

“The TRV-mediated virus-induced gene silencing (VIGS) was performed according to the previous method⁹⁶. For the construction of TRV2-CcOMT, a fragment of the *CcOMT1* ORF (300, 551-851bp) was amplified by PCR, which was then introduced into the TRV2 vector. The empty vector was used as a control. All the vectors were introduced into *A. tumefaciens* strain GV3101. The *Agrobacterium* cells carrying the TRV1 and TRV2 recombinants were grown in LB broth to OD600=1.0²⁷, and then *Agrobacterium* cultures were centrifuged and resuspended in infiltration buffer (10mM MgCl₂, 10mM MES, 200uM acetosyringone, 0.01% Tween20; pH 5.6). Resuspended *A. tumefaciens* TRV1 and TRV2-CcOMT1 were mixed in a 1:1 ratio and infiltrated into the citrus fruit on the trees, three single-tree-based biological replicates were set up⁹⁷. Citrus fruits were collected at two weeks after infiltration. Each treatment had three biological replicates with a minimum of 20 citrus fruits. These samples were analyzed for flavonoid content and gene expression as described above.”

The experiment figures of overexpression and silencing of *CcOMT1* were updated in Figure 6 as follows:

Figure 6. Transient overexpression, silencing and functional analysis of CcOMT1. a, Expression profiles of *CcOMT1* after infiltration with *CcOMT1*-pCAMBIA1301. b, Changes of 3,5,6,7,8,3',4'-heptamethoxyflavone after transient overexpression of *CcOMT1* in *C. chachiensis* peel. c, Relative expression of *CcOMT1* in virus-induced *CcOMT1* silencing *C. chachiensis* fruits. d, Changes in 3,5,6,7,8,3',4'-heptamethoxyflavone content in *C. chachiensis* fruits after virus-induced silencing of *CcOMT1*. CK, empty TRV2 vector control fruits.

2. Line 228, CZG-jg206 was proposed to be the main gene for the formation of bitter taste in *C. chachiensis*. Actually, this gene may not functional in relative to Cm1,2RhaT. The authors may need to confirm neohesperidoside by LC-MS and standards in *C. chachiensis*

Response: Thanks for your suggestions. A series of neohesperidoside compounds were detected in our metabolome data of all fruits and peels samples (detected by UPLC-MS/MS) including naringenin-7-O-neohesperidoside(naringin), diosmetin-7-O-neohesperidoside, poncirin, pinocembrin-7-O-neohesperidoside, and hesperetin-7-O-neohesperidoside (neohesperidin) (Table S7). Moreover, the expression of CZG_jg206 was highly correlated with these bitterness compounds such as naringin (cor=0.52, $P=0.04$) and poncirin (cor=0.65, $P=0.008$).

We updated the content in line 401 to 411 as follows:

“Besides, the candidate gene CZG_jg206 grouped with citrus bitterness-related gene Cm1,2RhaT (FLRT_CITMA) in group A. Cm1,2RhaT encodes a 1,2 rhamnosyltransferase, which is a key enzyme in the biosynthesis of the bitter flavanone-7-O-neohesperidosides of citrus⁴³, indicating CZG_jg206 might be the main gene for the formation of bitter taste in *C. chachiensis*. Several neohesperidosides were detected in young fruits, unmaturing, near mature and ripe peel of *C. chachiensis* by UPLC-MS/MS, including naringenin-7-O-neohesperidoside (naringin), diosmetin-7-O-neohesperidoside, poncirin, pinocembrin-7-O-neohesperidoside, and hesperetin-7-O-neohesperidoside (neohesperidin) (Table S7). Moreover, the expression level of CZG_jg206 was highly correlated with these bitterness compounds such as naringin (cor=0.52, $P=0.04$) and poncirin (cor=0.65, $P=0.008$; Figure S13).”

The corresponding figure S13 were showed as follows:

Figure S13 | The relative expression levels of CZG_jg206 (a) and the relative content of naringin (b) and poncirin (c). Error bars show standard errors, n=3.

3. UGT seems distracted the logic of this study. UGTs are associated with catalyzation of many metabolites. So the identified 96 UGTs are a pool of many kinds of function. Specifically, function of UGTs or specific UGTs for the PMFs of *C. chachiensis* need to be confirmed.

Response: Thank you so much for raising this point. We tried to illustrate the UGTs as many O-Glycoside derivatives of PMFs were detected in fruits and peel of *C. chachiensis*, which might be the important storage forms of PMFs in peels. Therefore, we searched UGT genes and predicted their functions in PMF production by phylogenetic analysis. Among the 96 candidate UGT genes, the expression level of 78 candidate UGT genes significantly showed correlations with the content of 29 PMFs (adjusted $P < 0.05$), indicating the key role of UGTs in modification of PMF. In order to make the content of the article more logical, we reorganized the section orders of manuscript content according to your suggestions and increased a section for glycosylation of PMFs and other flavonoids in *C. chachiensis*.

We updated the content in line 378 to 400 as follows:

“Glycosylation is catalyzed by UDP-glycosyltransferase (UGT), which is important for enriching the diversity of flavonoids. Several PMFs with glycosylation were identified in *C. chachiensis* fruits by UPLC-MS/MS, such as natsudaïdain-3-O-(5'-glucosyl-3-hydroxy-3-methylglutarate)glucoside, natsudaïdain-3-O-(3-hydroxy-3-methylglutarate)glucoside and 5,4'-dihydroxy-3,6,7,3'-tetramethoxyflavone-4'-O-glucoside. In addition, the relative content of natsudaïdain and its glycoside derivatives (natsudaïdain-3-O-(5'-glucosyl-3-hydroxy-3-methylglutarate)glucoside and natsudaïdain-3-O-(3-hydroxy-3-methylglutarate)glucoside) were highly correlated, indicating the presence of glycosylation modification of PMFs. PMFs might be metabolized into the O-Glycoside derivatives and stored stably in the peel. UGT might play a key role in this process. Therefore, we tried to investigate the potential UGT genes in *C. chachiensis*, especially the UGT genes correlated to the PMFs. Firstly, 96 candidate genes were identified by searching UGT genes against the whole genome of *C. chachiensis*. Then, a phylogenetic tree was constructed to define the phylogenetic relationship of the candidate UGT genes of *C. chachiensis* combined with the known UGT sequences extracted from *A. thaliana*, *C. sinensis*, *Zea mays*, *Catharanthus roseus*, *Pueraria lobata*, *Punica granatum*, *Ginkgo biloba*, and *O. sativa* (Figure S12). All UGT genes were clustered into 19 groups (Group A to R), and the correlation analysis showed that the expression levels of 78 candidate UGT genes were significantly correlated with the relative content of 29 PMFs (adjusted $P < 0.05$). Among these, CZG_jg11190 (Group A) had a higher correlation with natsudaïdain and its derivative

natsudaïdain-3-O-(3-hydroxy-3-methylglutarate) glucoside, which might be the possible gene for natsudaïdain glycosylation (cor=0.88, cor=0.75, p<0.05, respectively).”

4. The details for the metabolic data such as secondary mass spectrometry for substance qualitative identification, molecular ion fragment information. The instrument used for different purpose are different.

Response: Thanks for your comment. We are very sorry for the inaccurate description of the mass spectrometry results and have already revised the contents in line 326-329. The reactions were analyzed using HPLC/DAD and HPLC-QTOF/MS. The results showed that a new peak appeared on the chromatogram (Figure 6b). Compared with the reference standards, methyl-product displayed abundant [M+H]⁺ ions at m/z 433 of 1a in MS/MS, which was identified as 3,5,6,7,8,3',4'-heptamethoxyflavone.

Minor points

5. The sampling information of *C. chachiensis* from different places need to be clarified.

Response: Thanks for your comment. Peel samples of *C. chachiensis* were collected from six different orchards, including three orchards from core region (Tianma, Maichong, and Wubao) and three orchards from the other region (Taishan, Longmen and Daan) (Su, et al. 2023). We have clarified this information into line 487-489.

Table R1 the citrus orchard information from the core and other regions.

Region	Orchard	Location
Core region	Tianma	Shuangshui Town, Xinhui District, Jiangmen City, Guangdong Province, China
	Maichong	Shuangshui Town, Xinhui District, Jiangmen City, Guangdong Province, China
	Wubao	Shuangshui Town, Xinhui District, Jiangmen City, Guangdong Province, China
The Other region	Taishan	Taishan, Jiangmen city, Guangdong Province, China
	Longmen	Longmen county, Huizhou City, Guangdong Province, China
	Daan	Pingnan county, Guigang City, Guangxi Zhuang Autonomous Region, China

We also increase the clarifications for the sampling place in the method section in line 735 to 740 as follows:

“The transcriptome data of citrus peel from the core region and other region were produced in our previous study⁹⁰. Three representative orchards (named Tianma, Wubao and Maichong) in Xinhui (geo-authentic product region) of Guangdong province were selected as the core region. Three orchards such as Taishan from Jiangmen city, Longmen from Huizhou city of Guangdong province, and Daan from Guangxi province were selected to represent the non-core region.”

Reference:

Su, J. *et al.* Soil conditions and the plant microbiome boost the accumulation of monoterpenes in the fruit of *Citrus reticulata* ‘Chachi’. *Microbiome* 11, 61 (2023).

6. Line 54 to 62, it is not necessary to introduce the regulation of anthocyanin, the key point of this article is PMF.

Response: Thanks for your helpful suggestion. We have deleted the content related to the regulation of anthocyanin in the manuscript.

7. Line 206-207, 639-642, the sampling time were described as August, September, October, November, should be corrected as day after flowering (DAF).

Response: Thanks for your comment. We have corrected sample time of April, May, June, August, September, and November as 45 DAF, 75 DAF, 105 DAF, 165 DAF, 200 DAF, 230 DAF, and 260 DAF respectively. We also supplemented the sample figures and the responded DAF in Figure 3b.

Figure 3b. Different development stages of *C. chachiensis* fruits.

8. Line 10 and 89, please indicate whether Contig N50 or Scaffold N50? Contig N50 could represent genome quality better than Scaffold N50.

Response: Thanks for your comment. It is Scaffold N50, we have revised it as contig N50 as follows:

“Here, we report a high-quality chromosome-scale genome assembly of *C. chachiensis* where 99.36% of the 314.96 Mb genome sequence was assembled into 9 chromosomes with a contig N50 of 16.22 Mb.”

9. Line 142, Fig 2a can not reflect your result “*C. chachiensis* is related more closely to *C. clementina* than to *C. reticulata* (mangshan wild mandarin)”, maybe you intend to cite Fig 2c, please check.

Response: Thanks. We are sorry for this mistake. We have corrected it as Figure 2c.

10. *C. ichangensis* was one of the ancestors of Citrus, its position in Fig 2c clustering with *C. maxima* and *C. medica* is unusual. This node actually is somewhat sensitive to the data.

Response: Thanks for your comment. It might be caused by the high similarities of the citrus single-copy genes. We added five outgroup species into the phylogenetic analysis to reduce the number of single-copy genes. The position of *C. ichangensis* is located at the outgroup of *C. medica* and *C. maxima*, and all other citrus species except *P. trifoliata*. This result (Figure 2c) is consistent with previous studies (Wu, et al. 2018; Peng, et al. 2020). We updated the corresponding content in line 153 to 164 as follows:

“To investigate the evolution position of *C. chachiensis* in the genus *Citrus*, *C. chachiensis* genome was compared with 9 citrus and citrus-related species including *C. sinensis*, *C. clementina*, *C. reticulata*, *C. unshiu*, *C. maxima*, *C. medica*, *C. ichangensis*, *P. trifoliata*, and *Atalantia buxifolia*. Five outgroup species *Amborella trichopoda*, *Oryza sativa*, *Vitis vinifera*, *Arabidopsis thaliana*, and *Theobroma cacao* were also selected to construct the phylogenetic tree. Among the 10 *Citrus* species, a total of 13,474 gene families were shared and similar single-copy orthologs number was showed (Figure 2a and Figure 2b). 1,855 single-copy genes identified in all the 15 species were used to construct the phylogenetic tree (Figure 2c). *C. chachiensis* has a closer relationship to *C. clementina* than to *C. reticulata* (mangshan wild mandarin), which is consistent with the previous study that *C. clementina* belongs to a modern group and mangshan wild mandarin belongs to the group of older mandarins^{37,38}.”

We updated the figure 2c as follows:

Figure 2c. Phylogenetic tree of 15 plant species.

Reference:

Wu, G. A. *et al.* Genomics of the origin and evolution of *Citrus*. *Nature* 554, 311–316 (2018).

Peng, Z. *et al.* A chromosome-scale reference genome of trifoliate orange (*Poncirus trifoliata*) provides insights into disease resistance, cold tolerance and genome evolution in *Citrus*. *Plant J. Cell Mol. Biol.* 104, 1215–1232 (2020).

11. Line 156 and 158, the duplication event shared by all eudicot plants were WGT- γ , and Fig 2e did reflect the duplication event, but the red circles you marked were not apparent to demonstrate it, please refer to the Fig 2 of sunflower genome article (Badouin *et al.*, 2017).

Response: Thank you for raising this point. We have corrected it in Figure 2e as displayed in sunflower genome article.

The figure 2e was showed as follows:

Figure 2e. Dot plot of paralogs in the genome of *C. chachiensis*. The red circles indicate the ancient WGD event.

12. Line 153, it is an older WGD event, maybe you mean “ancient WGD event”.
Response: Thanks. We have corrected it as “ancient WGD event” in line 176.

13. Line 187 to 188, the reference of gene CICLE_v10026344mg should be added.
Response: Thanks. We had already added the reference of gene CICLE_v10026344mg that reported it with the O-methyltransferase activity on 6-OH, 8-OH and 3'-OH in line 210 of the revised manuscript.

Reference:

Liu, X. *et al.* Characterization of a caffeoyl-CoA O-methyltransferase-like enzyme involved in biosynthesis of polymethoxylated flavones in *Citrus reticulata*. *J. Exp. Bot.* 71, 3066–3079 (2020).

14. Line 216 PMF glycosylation? It was believed that this step was not happen in *C. chachiensis*.

Response: Thanks. Glycosylation is catalyzed by UDP-glycosyltransferase (UGT), which is important for enriching the diversity of flavonoids. Several PMFs with glycosylation were identified in *C. chachiensis* fruits by UPLC-MS/MS, such as natsudaïdain-3-O-(5'-glucosyl-3-hydroxy-3-methylglutarate)glucoside, natsudaïdain-3-O-(3-hydroxy-3-methylglutarate)glucoside and 5,4'-dihydroxy-3,6,7,3'-tetramethoxyflavone-4'-O-glucoside. In addition, the relative content of natsudaïdain and its glycoside derivatives (natsudaïdain-3-O-(5'-glucosyl-3-hydroxy-3-methylglutarate)glucoside and natsudaïdain-3-O-(3-hydroxy-3-methylglutarate)glucoside) were highly correlated, indicating the presence of glycosylation modification of PMFs. PMFs might be metabolized into the O-Glycoside derivatives and stored stably in peel. UGT might play a key role in this process. Therefore, we tried to investigate the potential UGT genes in *C. chachiensis*, especially the UGT genes correlated to the PMFs. We also revised the content in line 378 to 400.

15. Line 309 accumulated the highest PMF content followed by June, may need to confirm.

Response: Thanks for your suggestion. We are sorry for the inaccurate description. By compare the content of all PMFs in different development stages, we found that 16 of 29 PMFs show highest content in September (200DAF), 6 PMFs show highest content in June (105 DAF) and 2 PMFs show highest content in November (260 DAF) (Figure S7).

Therefore, we corrected the sentence into 430-432 as follows:

“Most of the PMFs exhibited the highest relative content in the peel of *C. chachiensis* at 200 DAF, followed by 105 DAF (Figure S7), suggesting that the gene expression and regulation of PMF biosynthesis could possibly be active during these stages.”

The Figure S7 was showed as follows:

Figure S7. The relative content of PMFs in *C. chachiensis* peel at three different stages. Compound name in red indicate the PMF reached its highest relative content at 200 DAF (a total of 16 PMFs show highest content at 200 DAF). Error bars show standard errors, n=3.

16. Figure S10, the unit for heptamethoxyflavone is irregular.

Response: Thanks for your comment. We had corrected the unit as $\text{mg}\cdot\text{g}^{-1}$ in the figure that updated as Figure 6.

17. Table S1 should have units.

Response: Thanks. We have added unit base pair (bp) for the length of chromosome in Table S1.

18. line 23 in the abstract, *C. clemintina* is not correct

Response: Thanks for your comment. We have corrected it as *C. clementina* in line 28.

Reviewer #2 (Remarks to the Author):

In this manuscript, the authors reported chromosome-level reference genome of *Citrus chachiensis*, the transcriptomic and metabolomic analysis towards exploring the regulatory networks and catalytic enzymes responsible for the biosynthesis of polymethylated flavonoids in *Citrus*, as well as the characterization of a O-methyltransferase that exhibits the detectable activity of 3-O-methylation of PMF in vitro. The manuscript was compiled with quite lots of data; however, unfortunately, the manuscript was poorly developed, and consequently its overall quality and readability is low. Many editorial and scientific issues pertaining to the text description, the experimental designs, and the data presentation can be found throughout the manuscript. For instance, the manuscript has cumbersome or redundant description, lacks structural integrity, contains scientific errors/inaccuracy, and has poor data presentation.

In the Introduction, it missed the information on the current research advance of *Citrus* genome sequencing, i.e., how many citrus species/varieties have been reported for their genomes and then the authors should link to their rationale conducting genome sequencing for *C. chachiensis*. In manuscript, the authors claimed that exploring *C. chachiensis* genome aimed to elucidate PMF biosynthesis and regulation. However, the authors also stated that PMFs are most abundantly enriched in the peel of *C. sinensis* and *C. reticulata*. Then it is not clear why the authors had to explore the genome of *C. chachiensis* but not those of *C. sinensis* and *C. reticulata*?

Response: Thanks for your constructive comments. We listed most recently sequenced citrus genomes including sweet orange (*Citrus sinensis*), pummelo (*Citrus maxima*), Citron (*Citrus medica*), clementine mandarin (*Citrus clementina*), mangshan wild mandarin (*Citrus reticulata*), Hong Kong kumquat (*Fortunella hindsii*), trifoliolate orange (*Poncirus trifoliata*), lemon (*Citrus limon*) and round lime (*Citrus australis*) (Nakandala et al., 2023; Bao et al., 2023; Peng et al., 2020; Zhu et al., 2019; Wang et al., 2018; Wang et al., 2017; Wu et al., 2014; Xu et al., 2013).

The reason why we explore the PMF biosynthesis and regulation of *C. chachiensis* is as follows:

Firstly, although PMFs exist almost exclusively in the citrus genus, particularly in the peel of sweet oranges (*C. sinensis*) and mandarin oranges (*C. reticulata*), the diversity and contents of PMFs in the *C. reticulata* were generally higher than those in the *C. sinensis* (Zhang et al., 2012).

Secondly, *C. chachiensis* is considered as one of cultivated varieties of *C. reticulata*, which has long been regarded as a medicinal mandarin. The dried and ripe peel of *C. chachiensis*, grown in Xinhui, Guangdong, China, named genuine Pericarpium Citri Reticulatae (Guangchenpi) was admired as the best Chenpi which possesses the excellent clinical efficacy (Fu et al., 2017; Sun et al., 2010). The main components of Guangchenpi are dietary flavonoids, which are divided into flavonoid glycosides and polymethoxylated flavonoids (PMFs). PMFs possess a wide range of bioactivities, including anticancer, neuroprotection, anti-inflammation, anti-obesity, antioxidant, antiatherosclerosis and so on (Chen et al., 2022; Falduto, et al., 2022; Jin et al., 2022; Liang et al., 2022; Md Idris et al., 2022; Wang et al., 2021; Zeng et al., 2020; Shajib et al., 2018). In particular, 3,5,6,7,8,3',4'-heptamethoxyflavone was examined with the anti-tumor-initiating activity, could be viewed as an effective candidate medicine for the treatment of cancer (Hirata et al., 2009; Sergeev et al., 2006; Iwase et al., 2001). Therefore, it is very important to study the biosynthesis and regulation mechanism of these PMFs for the future industrial application. Investigating *C. chachiensis* genome resources can help to understand the transcriptional regulation network and the biosynthetic pathway of the bioactive PMFs. Although, many citrus genomes have been assembled including sweet orange (*C. sinensis*), pummelo (*C. maxima*), clementine mandarin (*C. clementina*), mangshan wild mandarin (*C. reticulata*), Hong Kong kumquat (*F. hindsii*), trifoliolate orange (*P. trifoliata*), lemon (*C. limon*) and round lime (*C. australis*). The genome of *C. chachiensis* still has not been sequenced and interpreted.

Therefore, we assembled the genome of *C. chachiensis*, which provide a comprehensive genetic basis for deciphering the biosynthesis and regulation network of bioactive PMFs.

We updated the content in the introduction sections in line50 to 64 and 92 to line 107.

Reference:

- Xu, Q. et al. The draft genome of sweet orange (*Citrus sinensis*). *Nat. Genet.* 45, 59–66 (2013).
- Wu, G. A. et al. Sequencing of diverse mandarin, pummelo and orange genomes reveals complex history of admixture during citrus domestication. *Nat. Biotechnol.* 32, 656–662 (2014).

Wang, X. et al. Genomic analyses of primitive, wild and cultivated citrus provide insights into asexual reproduction. *Nat. Genet.* 49, 765–772 (2017).

Wang, L. et al. Genome of wild mandarin and domestication history of Mandarin. *Mol. Plant* 11, 1024–1037 (2018).

Zhu, C. et al. Genome sequencing and CRISPR/Cas9 gene editing of an early flowering Mini-Citrus (*Fortunella hindsii*). *Plant Biotechnol. J.* 17, 2199–2210 (2019).

Peng, Z. et al. A chromosome-scale reference genome of trifoliolate orange (*Poncirus trifoliata*) provides insights into disease resistance, cold tolerance and genome evolution in *Citrus*. *Plant J. Cell Mol. Biol.* 104, 1215–1232 (2020).

Bao, Y. et al. A gap-free and haplotype-resolved lemon genome provides insights into flavor synthesis and huanglongbing (HLB) tolerance. *Hortic. Res.* 10, uhad020 (2023).

Nakandala, U. et al. Haplotype resolved chromosome level genome assembly of *Citrus australis* reveals disease resistance and other citrus specific genes. *Hortic. Res.* 10, uhad058 (2023).

Zhang, J.Y. et al. Characterization of polymethoxylated flavonoids (PMFs) in the peels of ‘Shatangju’ mandarin (*Citrus reticulata* Blanco) by online high-performance liquid chromatography coupled to photodiode array detection and electrospray tandem mass spectrometry. *J. Agric. Food Chem.* 60, 9023–9034 (2012).

Fu, M. et al. Evaluation of bioactive flavonoids and antioxidant activity in Pericarpium Citri Reticulatae (*Citrus reticulata* ‘Chachi’) during storage. *Food Chem.* 230, 649–656 (2017).

Sun, Y. et al. Simultaneous determination of flavonoids in different parts of *Citrus reticulata* ‘Chachi’ fruit by high performance liquid chromatography—photodiode array detection. *Molecules* 15, 5378–5388 (2010).

Chen, P.Y. et al. 5-Demethylnobiletin inhibits cell Proliferation, downregulates ID1 expression, modulates the NF- κ B/TNF- α pathway and exerts antileukemic effects in AML cells. *Int. J. Mol. Sci.* 23, (2022).

Shajib, M. S. et al. Polymethoxyflavones from *Nicotiana plumbaginifolia* (Solanaceae) Exert Antinociceptive and Neuropharmacological Effects in Mice. *Front. Pharmacol.* 9, 85 (2018).

Md Idris, M. H. et al. Discovery of polymethoxyflavones as potential cyclooxygenase-2 (COX-2), 5-lipoxygenase (5-LOX) and phosphodiesterase 4B (PDE4B) inhibitors. *J. Recept. Signal Transduct. Res.* 42, 325–337 (2022).

Jin, Y.J. et al. in, Y. J. et al. Anti-obesity effects of polymethoxyflavone-rich fraction from Jinkyool (*Citrus sunki* Hort. ex Tanaka) leaf on obese mice induced by high-fat diet. *Nutrients* 14, (2022).

Falduto, M. et al. Anti-obesity effects of Chenpi: an artificial gastrointestinal system study. *Microb. Biotechnol.* 15, 874–885 (2022).

Wang, Y. et al. Tangeretin maintains antioxidant activity by reducing CUL3 mediated NRF2 ubiquitination. *Food Chem.* 365, 130470 (2021).

Liang, P.-L. et al. Three polymethoxyflavones from the peel of *Citrus reticulata* ‘Chachi’ inhibits oxidized low-density lipoprotein-induced macrophage-derived foam cell formation. *Front. Cardiovasc. Med.* 9, 924551 (2022).

Zeng, S.-L. et al. Citrus polymethoxyflavones attenuate metabolic syndrome by regulating gut microbiome and amino acid metabolism. *Sci. Adv.* 6, eaax6208 (2020).

Sergeev, I. N., Li, S., Colby, J., Ho, C.-T. & Dushenkov, S. Polymethoxylated flavones induce Ca²⁺-mediated apoptosis in breast cancer cells. *Life Sci.* 80, 245–253 (2006).

Hirata, T. et al. Identification and physiological evaluation of the components from citrus fruits as potential drugs for anti-obesity and anticancer. *Bioorg. Med. Chem.* 17, 25–28 (2009).

Iwase, Y. et al. Cancer chemopreventive activity of 3,5,6,7,8,3',4'-heptamethoxyflavone from the peel of citrus plants. *Cancer Lett.* 163, 7–9 (2001).

2) The following description (Lin 86-89) is scientifically wrong: “one of which is caffeoyl-CoA O-methyltransferase (COMT) with a wide range of substrates, including myoinositol, chalcones, caffeic acid, and so on. The other group is caffeoyl-CoA O-methyltransferase (CCoAOMT), which needs ion catalyzes in the reaction”. COMT represents caffeic acid 3/5-O-methyltransferase. Also, please check the grammar for the last sentence.

Response: Thanks so much for raising this point. We are very sorry for the incorrect definition of COMT and CCoAOMT. We have already made the following modifications. One of which is caffeic acid O-methyltransferase (COMT). The other subfamily is caffeoyl-CoA O-methyltransferase (CCoAOMT), which are usually cation dependent.

I revised the content from Line 81 to 84 as follows:

“Generally, there are two large groups of OMTs in plants. One group is caffeic acid O-methyltransferase (COMT), which has a wide range of substrates, including myoinositol, chalcones, and caffeic acid. The other group is caffeoyl-CoA O-methyltransferase (CCoAOMT), which needs ion catalyzes in the reaction^{23,24}.”

3) In the following descriptions: (Line 348-352) “AP2-ERF/ERF transcription factors were hub genes and strongly associated with the flavonoid biosynthesis genes (HCT: hydroxycinnamoyltransferase, FLS) and UDP-glycosyltransferase genes (UGTs), indicating that AP2-ERF/ERF transcription factors might modulate the flavonoid synthesis by regulating the HCT, FLS and UGT genes”; (Line 363-365)“AP2/ERF-ERF, UGT as well as HCT were identified as the key

gene in the regulatory network, which might drive the related gene expression and promoting the accumulation of PMFs”; (Line 441) “Early flavonoid biosynthesis genes such as HCT, PAL (phenylalanine ammonia-lyase), CHS, and CHI contributed to naringenin production...”, the authors claimed HCT is a key gene for flavonoid synthesis. This is not correct. HCT per se encodes an enzyme involved in monolignol synthesis and has nothing with flavonoid formation. It might be possible that some aromatic acyltransferases involved in the flavonoid glucoside modification but definitely they are not the ones involved in the early flavonoid biosynthetic steps.

Response: Thanks for your constructive comment. We re-construct the correlation network only using the differentially expressed transcription factors, flavonoid biosynthesis-related genes and the corresponding differential metabolites. We removed the gene of HCT. We found that transcription factors, such as AP2/ERF family, WRKY, and bZIP with the most degree showed significant correlations with flavonoid biosynthesis genes, such as chalcone isomerase (CHI) gene, flavonol synthase (FLS) gene, and flavanone-3-hydroxylase (F3H) gene. It is suggested that these transcription factors might regulate the expression level of flavonoid biosynthesis genes.

We revised the description in line 445 to 449 and the corresponding figure was updated in Figure S16.

Figure S16. The correlation network of DEGs and DMs between 105 DAF and 200 DAF. Squares indicate differentially accumulated metabolites (DM); triangles indicate differentially expressed transcription

factors; green circles indicate differentially expressed flavonoid biosynthesis genes; gray dash represent the positive correlation ($r > 0.8$, $p < 0.05$).

We also revised the other corresponding content as “All PMFs are derived from the main flavonoids naringenin (Figure 8). Early flavonoid biosynthesis genes such as CHS, and CHI contributed to naringenin production. The late flavonoid biosynthesis genes such as FLS, F3H, and UGT were responsible for the modification of naringenin, producing different types of naringenin derivatives. These chemical products could furthermore undergo glycosylation and methylation, where UGT and OMT played an essential role, respectively. After that, PMFs with different moieties could be produced. Transcription factors such as AP2/ERF-ERF, HB-HD-ZIP, LIM, and WRKY might be play a significant role in this process.” in line 497 to 504.

4) Regarding metabolomic analysis, the authors described in the Results that the metabolite extracts were analyzed by GC-MS; however, in the Method the LC-MS methods were described.

Response: Thanks for your comment. We detected the metabolites using LC-MS methods. We are sorry for this written error and had corrected it as “UPLC-MS/MS” in line 242.

5) The authors reported that metabolomic study revealed 94 flavones, 75 flavonols, 49 flavonoid carbonoside, 49 flavanones, 14 tannins, 12 chalcones, 4 dihydroflavonols and others. It is better also present the detail metabolites list and their chemical assignments in a supplemental datasheet or the figures, and be cited within the main text. In addition, the method on chemical identification/assignments of these detected flavonoids should be provided in the Method section.

Response: Thanks for your helpful suggestion. We have presented the detail of this metabolite into Table S7 and Figure S6. The corresponding method was described in line 749 to 763 as follows:

“All the above 15 samples including fruits and peel were frozen and dried in a freezer separately before grinding into powder by the grinder for 90s at 30Hz. Then 100mg powder of each sample was dissolved in 1.2 mL of 70% methanol extract. The mixture was vortexed for 30s and placed 30 min and repeated this operation for 6 times. Then the mixture of each sample was placed in a refrigerator at 4°C overnight. Next, each mixture was centrifugated for 10 min at 12,000 rpm and the supernatant was filtered by 0.22 μm pore size microporous membrane. The filtrate was transferred into the detection bottle and used for UPLC-MS/MS analysis. The analyzer system included Ultra Performance Liquid Chromatography (UPLC) SHIMADZU Nexera X2, Kyoto, Japan) and Tandem Mass Spectrometry (MS/MS) (Applied Biosystems 4500 QTRAP, AB Sciex, Framingham, MA, USA). Qualitative analysis of the compounds was

performed based on a self-construction database and quantitative analysis relied on multiple reaction monitoring (MRM) mode. Analyst 1.6.3 (<https://sciex.com/products/software/analystsoftware>) was used for processing the mass spectrometry data.”

6) The exploration of CcOMT1 in the context surprisingly disconnected from the described transcriptomic and metabolic studies. The authors stated that CcOMT1 was identified from protein database by sequence similarity searching with a tomato flavonoid 3-OMT, ShMOMT3, which appears that the CcOMT1 characterization was conducted parallelly with the described omics studies, and those different studies were just simply compiled together, but lack logical connection and integration.

Response: Thanks for your constructive comment. We have reorganized the context and made the content logically coherent. The relative contents of natsudaïdain and 3,5,6,7,8,3',4'-heptamethoxyflavone were highly correlated ($r=0.628$, $P=0.012$), indicating the possible production of 3,5,6,7,8,3',4'-heptamethoxyflavon via natsudaïdain. Therefore, we hypothesized that 3,5,6,7,8,3',4'-heptamethoxyflavone might be transformed from the precursor compound natsudaïdain, by an OMT adding the O-methoxy group into the 3-OH site. Then we use the sequence of ShMOMT3, a 3-OMT of flavonoids in tomatoes⁴⁰ to search against the assembled-protein database of *C. chachiensis*. Several candidate genes with high similarities were obtained from the BLAST search including CZG_jg11962 (52%), CZG_jg11963 (51%), CZG_jg17822 (51%), CZG_jg18299 (51%), CZG_jg23745 (50%) and CZG_jg23753 (50%). The gene-specific primers were designed to obtain candidate genes from the cDNA library of *C. chachiensis* leaves. All these candidate genes were cloned and tested. Interestingly, only the protein encoded by gene CZG_jg17822 that had been predicted to perform methylation on 3'-position displayed the expected function (catalyze on 3-hydroxyl group) and was named as CcOMT1.

The corresponding content was in line 303 to 313.

7) Regarding the characterization of CcOMT1, a few expected experiments were missing:

To determine a catalytic enzyme and compare it with the others (or its mutant variants), the kinetic parameters should be measured under the apparent optimal conditions; moreover, since phenolic OMTs often exhibit promiscuity to their substrates and with regio-specificity, the enzymatic assays of CcOMT1 should be conducted with a broader phenolics, including those bearing the hydroxy moiety at different positions.

Response: Thanks for your constructive comments. We agree with this reviewer's comment. We had already carried out a set of experiments to determine the kinetic parameters of

CcOMT1. The Michaelis–Menten constant (K_m) were determined from non-linear regression fitting, and the turnover number (K_{cat}) were calculated (Figure S11). Then, we compared the activity of several OMTs that catalyze flavonoids in Table S11. Although all those enzymes catalyzed flavonoids, they catalyzed different substrates. The K_{cat}/K_m value indicated that CcOMT1 had low catalytic activity in the 10 OMTs. Therefore, site-specific mutation based on semi-rational design was performed on CcOMT1 to improve its catalytic activity. Variant CcOMT1-W141Y with the highest enzyme activity was selected for further study. As shown in Figure S11, compared with CcOMT1, the catalytic efficiency of CcOMT1-W141Y on natsudaïdain has been significantly improved.

We also conducted a new set of experiments to examine the substance promiscuity of CcOMT1. The results indicated that CcOMT1 could catalyze not only the 3-hydroxyl group of flavonoids, but also the 5-, 7-, 3'-, 4'- hydroxyl groups of flavonoids, and the catalytic reaction could not be carried out when the methoxy group was nearby (Figure S10).

We updated the content in line 337 to 342 and line 330 to 336, respectively.

Table S11 Function of flavonoids related OMTs

OMT gene	Native Species	GenBank	Substrate	K_m (μM)	K_{cat} (s^{-1})	K_{cat}/K_m ($\text{s}^{-1}\mu\text{M}^{-1}$)	Reference (PMID)
ObFOMT2	Sweet basil (Ocimum basilicum L.)	JQ653276	scutellarein	0.25	0.029	0.12	Berim et al., 2012 (22923679)
CCoAOMT7	Arabidopsis thaliana	At4g26220	Eriodictyol	63	0.08	1.3×10^{-3}	Wils et al., 2013 (23416302)
POMT-7	Poplar (Populus deltoids)	TC29789	Isorhamnetin	26.6	0.03	1.1×10^{-3}	Kim et al., 2008 (18817819)
MpalOMT3	Marchantia paleacea	MZ161832	Myricetin	47.6	0.0297	6.2×10^{-4}	Xu et al., 2022 (34895539)
PaCCoAOMT2	Polypodiodes amoena	MK164418	Quercetin	91.59	0.05	5.3×10^{-4}	Zhang et al., 2019 (30685696)
CsCCoAOMT1	Citrus sinensis	XM_006486982.4	eriodictyol	56.9	5.06×10^{-3}	8.9×10^{-5}	Liao et al., 2023 (37086474)
Vp-OMT4	Vanilla planifolia	JF344740	tricetin	158	0.0111	6.6×10^{-5}	Widiez et al., 2011 (21629984)
CrOMT1	Citrus reticulata	ESR39161	Kaempferol	248.1	12.4×10^{-4}	5.0×10^{-6}	Liu et al., 2020 (32182355)
CcOMT1-W141Y	C. chachiensis	-	natsudaïdain	20.8	5.7×10^{-7}	2.7×10^{-8}	This work
CcOMT1	C. chachiensis	-	natsudaïdain	25.4	3.7×10^{-7}	1.4×10^{-8}	This work

Figure S11. Determination of kinetic parameters for CcOMT1 (A) and the mutant CcOMT1-W141Y (B). The K_m value of CcOMT1 and CcOMT1-W141Y for natsudaïdain (1) are $25.4 \pm 11.3 \mu\text{M}$ and $20.8 \pm 7.6 \mu\text{M}$, respectively. The K_{cat}/K_m value of CcOMT1 and CcOMT1-W141Y on natsudaïdain (1) are $1.4 \times 10^{-8} \text{ s}^{-1} \mu\text{M}^{-1}$ and $2.7 \times 10^{-8} \text{ s}^{-1} \cdot \mu\text{M}^{-1}$, respectively.

Figure S10. Substrate promiscuity of CcOMT1. A) The yields of methylated products catalyzed by CcOMT1. B) Structures of substrates for CcOMT1.

8) The CcOMT1 expression pattern should be highlighted in a supplemental figure.

Response: Thanks for your comment. We have added *CcOMT1* expression pattern into Figure S5 as follows:

Figure S5. Relative expression levels of a COMT gene CZG_jg17822 (CcOMT1) in different development stages and tissues. Error bars show standard errors, $n \geq 3$.

9) Line 30-31, “CcOMT1 could catalyze natsudaïdain to produce 3,5,6,7,8,3’,4’-heptamethoxyflavone by adding a methyl donor SAM”. What does it mean “adding a methyl donor SAM”?

Response: Thanks for your comments. We are very sorry for not clearly describing our experimental process. CcOMT1 could transfer a methyl group provided by SAM into 3-hydroxyl of natsudaïdain, forming the expected product 3,5,6,7,8,3’,4’-heptamethoxyflavone. We revised the sentence in line 32 to 35.

10) Line 746-747, “The biochemical properties of CcOMT1 were studied using natsudaïdain as a sugar receptor and SAM as a sugar donor”---what do they mean “sugar receptor” and “sugar donor”?? CcOMT is a methyltransferase not a glucosyltransferase, and SAM is a methyl donor.

Response: Thanks for your comments. We are very sorry for the mistake. We had already made the following amendments. The biochemical properties of CcOMT1 were studied using natsudaïdain as a methyl receptor and SAM as a methyl donor. We revised the sentence in line 813 to 814.

11) The sentence needs to be rephrased “the limits of knowledge is to know the transcriptional regulation network and the biosynthetic pathway of PMF”

Response: Thanks for your comment. We have re-organized the content in this paragraph and revised this sentence as follows:

“Despite the increasing interest in “Guangchenpi”, there is still a limited understanding about the molecular mechanism regulating the biosynthesis of PMFs in citrus fruit.”

12) “illuminate the key OMTs”----might be better to change it as “elucidate the key OMTs”

Response: Thanks for your comment. We have changed it in line 85.

13) The sentence (line 110—112) “The estimated genome size of *C. chachiensis* was predicted to be 284.29 Mb with 1.01% heterozygosity ratio based on the analysis of k-mer depth distribution by short-read data (Figure 1a; Figure S1).” was pointed to the Figure 1a. However, the figure 1a shows nothing with *C. chachiensis* genome.

Response: Thanks for your comment. We have corrected it as Figure 1b.

14) Line 246-247, “These CGT genes exhibited higher expression levels in fruits and peel, which would contribute to higher content of corresponding products in fruits and peel”. The CGT gene expression data should be presented to support the statement.

Response: Thanks for your constructive comment. We have added this information into Figure S14 as shown below:

Figure S14. The relative expression levels of the candidate CGT gene CZG_jg8991. Error bars show standard errors, n≥3.

15) The description similar to the line 384-386 “3,5,6,7,8,3',4'-heptamethoxyflavone is one of the most important PMFs with multiple medicinal effects....” is repeated several times in the text.

Response: Thanks for your comment. We have deleted the repeated description and revised them in line 60-62 and in line 295 to 296.

16) Line 343—347, “The MEyellow module was highly correlated with near mature peel in September ($r=0.89$, $P=9e-06$) and significantly positive correlated with PMFs ($r=0.7$, $P=0.004$) and flavonoid C-glycoside ($r=0.8$, $P=3e-04$). The MEblue module was correlated with young fruits in April and negatively correlated with flavonols ($r=-0.7$, $P=0.004$) and flavonoid-O-glucosides ($r=-0.67$, $P=0.006$)”, this description should be accompanied with a figure presentation.

Response: Thanks for your constructive comment. We have added this information into Figure S17 and Figure S18 as follow:

Figure S17. The relationship between modules and fruit development stages.

Figure S18. The relationship between modules and the relative content of different kinds of flavonoids.

17) The uploaded main figures have so low resolution and their contents are barely observable.

Response: Thanks for your comment. We have displayed all the figures with high resolutions and updated them in the manuscript.

Reviewer #1 (Remarks to the Author):

The authors provided details and addressed most of the questions I concerned. Two key questions are necessary to be addressed to support the conclusions from this study, and to meet the high-level standards from NC journal.

1. The function of the key gene COMT1. VIGS assay are a kind results of transient assay, not from stable gene transformation assay. Moreover, as showed in the Table S11, the enzymatic activities of CcOMT1 and its mutant are extremely low, and the substrate-product transfer coefficient were listed in the MS as K_{cat} , they are also negligible, which means the CcOMT1 is not a main gene for the product. It is also possible that other strong OMTs functionalized along with it to alter the contents of PMFs in transient assay, by means of complex or metabolon, which is far from explained in the MS. Thus, the results obtained now are by no means sufficient to support the conclusion. Strong evidence such as stable transformation evidence are necessary to support this conclusion.

2. All the reviewers raised the concern on the details of all main flavonoids identified in this study. On secondary mass spectrometry for substance qualitative identification, molecular ion fragment information are necessary to be provided. Because the info. along with authentic standard identification, regio-substitution law, chemical bond breaking law are very important to judge which metabolite is present or absent in the species the author studied.

3. The identification of the presence of neohesperidosides and the key gene need to double check and more functional evidence. the main 12RhaT is nonfunctional in loose-skin mandarins due to SNP or frame-shifting, and other genes with 12RhaT function were almost not expressed. Thus, the presence of the neohesperidosides should be barely detected, therefore the function of CZG-jg206 as 12RhaT should not be possible. Finally, if the authors want to argue on the issue, pls provide data from in vitro enzyme characterization or functional evidence, correlation analysis is not enough and may mislead the authors.

Reviewer #2 (Remarks to the Author):

The resubmission by Wen et al provided additional experimental data and re-organized context; by which substantial improvement was made for the readability and clarity of the manuscript. The authors have addressed the most of my concerns.

I feel it might be better that the NC editorial office staff could help to further work on the English writing of the manuscript and to make more concise description/presentation.

A few minor issues:

1. Line 84 "(CCoAOMT), which needs ion catalyzes in the reaction" might change to "which needs ionic cofactors in the reaction"

2. please note that O-methyltransferase is to transfer methyl group to the hydroxyl moiety of substrate, it is not to add the "methoxyl" group. Such error occurs throughout the manuscript, for examples,

Line 213, "3'-OH methoxylation"

Line 304, "by an OMT adding the O-methoxy group into"

Line 545, "can directly add the O-methoxy group to the 3-OH site"

3. In addition, O should be italic in O-methylation or O-glycosylation, O-methyltransferase/-glucosyltransferase, and in the compound names.

Dear Editor and Reviewers,

We greatly appreciate your constructive comments and suggestions. We have performed additional analysis and experiments to strengthened all of our major conclusions in the manuscript. Below we provide comments on each of their suggestions or concerns.

Reviewer #1 (Remarks to the Author):

The authors provided details and addressed most of the questions I concerned. Two key questions are necessary to be addressed to support the conclusions from this study, and to meet the high-level standards from NC journal.

- 1. The function of the key gene COMT1. VIGS assay are a kind results of transient assay, not from stable gene transformation assay. Moreover, as showed in the Table S11, the enzymatic activities of CcOMT1 and its mutant are extremely low, and the substrate-product transfer coefficient were listed in the MS as K_{cat} , they are also negligible, which means the CcOMT1 is not a main gene for the product. It is also possible that other strong OMTs functionalized along with it to alter the contents of PMFs in transient assay, by means of complex or metabolon, which is far from explained in the MS. Thus, the results obtained now are by no means sufficient to support the conclusion. Strong evidence such as stable transformation evidence are necessary to support this conclusion.**

Response: Thanks for your comment. In order to validate the functions of CcOMT1, we had tested with various substrates, including natsudaidain, 3'-demethylnobiletin, wogonin, genistein, and isorhamnetin as mentioned in our previous response. Our findings indicate that CcOMT1 exhibits multiple-site activities, catalyzing not only the 3-hydroxyl group of flavonoids, but also the 5-, 7-, 3'-, 4'- hydroxyl groups of flavonoids (Figure S10).

While CcOMT1 displayed the third-highest relative activity on natsudaidain among these substrates and exhibited a extremely low K_{cat} , it is noteworthy that CcOMT1 is the sole OMT from *C. chachiensis* capable of transferring a methyl group to 3-OH of natsudaidain among all of the candidate genes based on our BLAST search results, including CZG_jg11962 (52%), CZG_jg11963 (51%), CZG_jg17822 (51%), CZG_jg18299 (51%), CZG_jg23745 (50%) and CZG_jg23753 (50%) which used ShMOMT3, a 3-OMT of flavonoids in tomatoes as a reference. Only CcOMT1 (CZG_jg17822) showed the expected function.

It is the first time to discover that an OMT could produce 3,5,6,7,8,3',4'-heptamethoxyflavone (HPMF) directly from a PMF substrate, natsudaidain (3-hydroxy-3',4'5,6,7,8-hexamethoxyflavone). This distinguishes our study from previous researches that demonstrated high K_{cat} and k_{cat}/K_m on non-polymethoxylated flavonoid substrates including apigenin, luteolin, myricetin, quercetin and kaempferol (refer to the updated Table S11) . Moreover, the products of these reactions were not polymethoxylated flavonoids (with more than three methoxy groups). Considering the increased difficulty of reactions with more methoxy groups on PMFs because of steric hindrance (Wang et al., 2018), we

believe that the K_{cat} and k_{cat}/K_m values of CcOMT1 on natsudaidain cannot be directly compared with other OMTs.

Another significant factor contributing to the remarkably low activities of both CcOMT1 and its mutant on natsudaidain may be related to the content of the target compound HPMF in *C. chachiensis*. The content of HPMF in the fruit peel of *C. chachiensis* is extremely low. According to the findings by Peng et al., the content of HPMF in 60 days post anthesis fruit flavedo of *C. chachiensis* is only $0.75 \text{ mg}\cdot\text{g}^{-1} \text{ DW}$, while the precursors of HPMF, such as nobiletin and tangerine, are $20 \text{ mg}\cdot\text{g}^{-1} \text{ DW}$ and $11 \text{ mg}\cdot\text{g}^{-1} \text{ DW}$, respectively (Peng et al., 2021). The content of HPMF is only 3.75% of nobiletin and 6.82% of tangerine. Additionally, the content of HPMF was not detected in the leaves of *C. chachiensis* (Peng et al., 2021). Our results in Figure 6b and 6d also demonstrated that the content of HPMF is extremely low, approximately $0.20 \text{ mg}\cdot\text{g}^{-1} \text{ FW}$ in fruits. Therefore, all these results suggested that HPMF was trace ingredient in *C. chachiensis*, indicating the O-methyltransferases related to HPMF biosynthesis should exhibit low catalytic activity on the precursor compounds of HPMF *in vivo* of the plant. This observation aligns with our experimental finding that CcOMT1 is a crucial OMT in *C. chachiensis* responsible for producing HPMF. Furthermore, it is essential to clarify that CcOMT1 is a multifunctional enzyme capable of catalyzing multiple activity sites on flavonoids, including the 3-, 5-, 7-, 3'-, and 4'-hydroxyl groups of flavonoids.

Finally, the process of generating stably transformed citrus plants is both arduous and time-consuming, with transformation efficiencies often being low (Conti et al., 2021). Due to the unique accumulation of PMFs, which is exclusive to mandarins and mandarin hybrids (Peng et al., 2021), it is impractical to validate the function of CcOMT1 in transgenic transgenic *Arabidopsis* plants. The VIGS assay is frequently used for validating gene functions in citrus and other non-model plants in numerous studies (Wu et al., 2023; Rössner et al., 2022; Zhao et al., 2021; Dai et al., 2018; Chantreau et al., 2015; Deng et al., 2012). Moreover, the function of CcOMT1 catalyzed natsudaidain to HPMF was also validated through *in vitro* experiment (Figure 5).

In summary, our study marks the first discovery of an OMT capable of directly producing 3,5,6,7,8,3',4'-heptamethoxyflavone (HPMF) from a PMF substrate, natsudaidain (3-hydroxy-3',4',5,6,7,8-hexamethoxyflavone). This distinguishes our findings from previous studies that focused on enzymes with high K_{cat} and K_{cat}/K_m values functioning on non-polymethoxylated flavonoid substrates, including apigenin, luteolin, myricetin, quercetin, and kaempferol, with the products not being PMFs. CcOMT1 is identified as a multifunctional enzyme with the capacity of producing HPMF. The lower efficiency of CcOMT1 on natsudaidain aligns with the extremely lower content of the target compound HPMF in *C. chachiensis* itself.

We revised the corresponding content in line 361 to 372:

“However, the low enzymatic activities of both CcOMT1 and its mutants on natsudaidain may be related to the extremely low content of the target compound HPMF in *C. chachiensis*. A recent study have reported that the content of HPMF is as low as $0.75 \text{ mg}\cdot\text{g}^{-1} \text{ DW}$ in 60 days post anthesis fruit flavedo of *C. chachiensis*, while the precursors of HPMF,

such as nobiletin and tangerine, are $20 \text{ mg}\cdot\text{g}^{-1} \text{ DW}$ and $11 \text{ mg}\cdot\text{g}^{-1} \text{ DW}$, respectively⁴³. The content of HPMF is only 3.75% of nobiletin and 6.82% of tangerine. Our *in vivo* experiments results below also demonstrated that the content of HMPF in the control groups is extremely low, approximately $0.20 \text{ mg}\cdot\text{g}^{-1} \text{ FW}$ in fruits of *C. chachiensis* (Figure 6b left and 6d left). Therefore, it was suggested that HPMF was trace ingredient in *C. chachiensis*, indicating the *O*-methyltransferases related to HPMF biosynthesis should exhibit low catalytic activity on the precursor compounds of HPMF *in vivo* of the plant.”

Figure S10. Substrate promiscuity of CcOMT1. A) The yields of methylated products catalyzed by CcOMT1. B) Structures of substrates for CcOMT1.

Figure 6. Transient overexpression, silencing and functional analysis of CcOMT1. a, Expression profiles of CcOMT1 after infiltration with CcOMT1-pCAMBIA1301. b, Changes of 3,5,6,7,8,3',4'-heptamethoxyflavone (HPMF) after transient overexpression of CcOMT1 in *C. chachiensis* peel. The effect of transient expression of CcOMT1 on the content of the HPMF in the peel of *C. chachiensis* was measured after infiltration with CcOMT1-PCAMBIA1301, with empty vector as control. c, Relative expression of CcOMT1 in virus-induced CcOMT1 silencing *C. chachiensis* fruits. d, Changes in HPMF content in *C. chachiensis* fruits after virus-induced silencing of CcOMT1. CK, empty TRV2 vector control fruits; VIGS, virus-induced CcOMT1 gene silencing fruits. The content of flavonoids is expressed as mg.g⁻¹ FW. Data are presented as mean \pm standard error from three independent biological replicates and Student's t-test was used for statistical analyses compared with corresponding control. Significant differences are indicated with asterisks above the bars (*, $P < 0.05$; **, $P < 0.01$; and ***, $P < 0.001$).

Table S11. The Function of flavonoids related OMTs.

Enzyme	Group	Substrate	K _m (μM)	k _{cat} (s ⁻¹)	k _{cat} /K _m (μM ⁻¹ s ⁻¹)	species	References
POMT7	F7OMTs	Luteolin	26.4	0.07	2.5 × 10 ⁻³	Poplar	Kim, et al., 2008
		Chrisoeryol	30.3	0.11	3.5 × 10 ⁻³		
		Quercetin	28.7	0.06	2.2 × 10⁻³		
		Isorhamnetin	26.6	0.03	1.1 × 10 ⁻³		Joe, et al., 2010
		Kaempferol	30.6	0.05	1.5 × 10⁻³		
		Quercetin	29.8	0.071	2.4 × 10⁻³		
ObFOMT1	F7OMTs	Kaempferol	32.4	0.064	1.9 × 10⁻³	Sweet Basil	Berim, et al., 2012
		Apigenin	0.032	0.091	2.81		
ObFOMT2	F6/4' OMTs	- Luteolin	0.24	0.10	0.43	Sweet Basil	Berim, et al., 2012
		- Scutellarein	0.58	0.047	0.08		
		S-adenosylmethionine	2.5	-	-		
ObFOMT3	F6/4' OMTs	Apigenin	0.059	0.043	0.73	Sweet Basil	Berim, et al., 2012
		Luteolin	0.25	0.049	0.20		
		Scutellarein	0.25	0.029	0.12		
ObFOMT4	F6/4' OMTs	S-adenosylmethionine	1.9	-	-	Sweet Basil	Berim, et al., 2012
		Scutellarein-7-methyl ether	0.41	0.085	0.21		
		Cirsimaritin	0.042	0.098	2.32		
ObFOMT5	F6/4' OMTs	Genkwanin	0.13	0.057	0.44	Sweet Basil	Berim, et al., 2012
		S-adenosylmethionine	36	-	-		
		Scutellarein-7-methyl ether	0.098	0.013	1.28		
ObFOMT6	F6/4' OMTs	Ladanein	0.054	0.014	2.55	Sweet Basil	Berim, et al., 2012
		S-adenosylmethionine	21	-	-		
		Scutellarein-7-methyl ether	0.036	0.066	1.81		
StF3OMT	F3OMTs	Cirsimaritin	0.087	0.05	0.58	Sweet Basil	Berim, et al., 2012
		Genkwanin	0.13	0.072	0.55		
		S-adenosylmethionine	41	-	-		
ShMOMT3	F3OMTs	Scutellarein-8-methyl ether	0.11	0.041	0.039	Sweet Basil	Berim, et al., 2012
		Ladanein	0.036	0.029	0.8		
		S-adenosylmethionine	78	-	-		
PaF6OMT	F6OMTs	Kaempferol	13	-	-	Serratula tinctoria	Haug, et al., 2004
		Myricetin	7	-	-		
		Myricetin	0.55	0.82	1.49		
Pa4'OMT	F4'OMTs	3-Methyl myricetin	2.07	1.66	0.93	Solanum habrochaites	Schmidt, et al., 2012
		7-Methyl quercetin	9.98	0.39	0.05		
		SAM	10.86	2.94	0.29		
ShMOMT1	F3/5'OMTs	Baicalein	43.21	0.039	9.12 × 10 ⁻¹	livenwort	Zhang, et al., 2016
		Scutellarein	37	0.039	1.05 × 10 ⁻³		
		Apigenin	31	0.117	3.76 × 10 ⁻³		
ShMOMT2	F7/4'OMTs	Luteolin	52.1	0.08	1.54 × 10 ⁻³	Plagiochasma appendiculatum	Liu, et al., 2017
		Myricetin	0.46	1.59	3.46		
		3-Methyl myricetin	0.21	0.45	2.14		
CCoAOMT7	F3/5'OMTs	SAM	16.64	0.47	0.03	Solanum habrochaites	Schmidt, et al., 2011
		Myricetin	1.88	7.40 × 10 ⁻³	4.41 × 10 ⁻³		
		4'-Methylkaempferol	2.27	5.76 × 10 ⁻³	2.53 × 10 ⁻³		
FAOMT	F3/5'OMTs	7-Methylquercetin	2.3	6.40 × 10 ⁻³	2.78 × 10 ⁻³	Solanum habrochaites	Schmidt, et al., 2011
		SAM	18.71	1.64 × 10 ⁻²	8.75 × 10 ⁻⁴		
		Eriodictyol	63	0.08	1.3 × 10 ⁻³		
CrOMT2	Multiple-Site (F7/3'/5' OMTs)	Luteolin	5	0.026	5.2 × 10 ⁻³	Arabidopsis thaliana	Wils, et al., 2013
		Quercetin 3-O-glucoside	7.6	2.38	0.31		
		Quercetin 3-O-glucoside	2.7	0.9	0.33		
CcOMT1-W141Y	Multiple-Site (F3/5'/3'/4' OMTs)	Luteolin	7.6	20.5 × 10 ⁻³	2.7 × 10 ⁻³	Vitis vinifera cv. Cabernet Sauvignon	Lücker, et al., 2010
		Tricetin	8.7	21.2 × 10 ⁻³	2.45 × 10 ⁻³		
		Baicalein	6.5	3.9 × 10 ⁻³	5.99 × 10 ⁻⁴		
CcOMT1	Multiple-Site (F3/5'/3'/4' OMTs)	Quercetin	2.9	2.6 × 10 ⁻³	9.14 × 10 ⁻⁴	Citrus reticulata cv. Ougan	Liu, et al., 2020
		Myricetin	15.4	15 × 10 ⁻³	9.80 × 10 ⁻⁴		
		Eriodictyol	4.6	7.6 × 10 ⁻³	1.65 × 10 ⁻³		
CcOMT1	Multiple-Site (F3/5'/3'/4' OMTs)	Caffeic acid	31.2	16.3 × 10 ⁻³	5.23 × 10 ⁻⁴	C. chachiensis	This work
		Natsudaidain	25.4	3.7 × 10 ⁻⁷	1.4 × 10 ⁻⁸		

* "-" means unprovided

Reference:

- [1] Chantreau, et al. Functional analyses of cellulose synthase genes in flax (*Linum usitatissimum*) by virus-induced gene silencing. *Plant Biotechnol J.* 2015;13(9):1312-1324.
- [2] Conti, et al. Citrus Genetic Transformation: An Overview of the Current Strategies and Insights on the New Emerging Technologies. *Front. Plant Sci.* 2021;12, 768197.
- [3] Dai, et al. FcWRKY40 of *Fortunella crassifolia* functions positively in salt tolerance through modulation of ion homeostasis and proline biosynthesis by directly regulating SOS2 and P5CS1 homologs. *New Phytol.* 2018;219(3):972-989.
- [4] Deng, et al. Virus-induced gene silencing for Asteraceae--a reverse genetics approach for functional genomics in *Gerbera hybrida*. *Plant Biotechnol J.* 2012;10(8):970-978.

- [5] Peng, et al. Comparative profiling and natural variation of polymethoxylated flavones in various citrus germplasms. *Food Chem.* 2021;354:129499.
- [6] Rössner, et al.. VIGS Goes Viral: How VIGS Transforms Our Understanding of Plant Science. *Annu Rev Plant Biol.* 2022;73:703-728.
- [7] Wu et al. CaMYB12-like underlies a major QTL for flavonoid content in pepper (*Capsicum annuum*) fruit. *New Phytol.* 2023;237(6):2255-2267.
- [8] Wang et al. Bioactive flavonoids in medicinal plants: Structure, activity and biological fate. *Asian J Pharm Sci.* 2018;13(1):12-23.
- [9] Zhao, et al. (2021). Three AP2/ERF family members modulate flavonoid synthesis by regulating type IV chalcone isomerase in citrus. *Plant Biotechnol J*, 19(4), 671–688.

2. All the reviewers raised the concern on the details of all main flavonoids identified in this study. On secondary mass spectrometry for substance qualitative identification, molecular ion fragment information are necessary to be provided. Because the info. along with authentic standard identification, regio-substitution law, chemical bond breaking law are very important to judge which metabolite is present or absent in the species the author studied.

Response: Thanks for your comment. We have provided the ion current diagrams (Figure R1), the ion current overlapped diagrams (Figure R2), the MRM metabolite detection multi-peak diagram (Figure R3) and integral correction diagram for quantitative analysis (Figure R4) below. We also added these figures into the updated supplementary figure file (refer to Extended data Figure R1 to R4).

Figure R2 shows the reproducibility of metabolite extraction and detection, such as technical duplication. This is achieved by analyzing the overlapping display of total ion flow plots (TIC plots) obtained from mass spectrometry detection of different quality control (QC) samples. QC samples are prepared from a mixture of sample extracts.

The MRM metabolite detection multi-peak diagrams (Figure R3) showed all the compounds detected in the samples. Each mass spectrum peak color representing one detected metabolite. The characteristic ions of each compound were selected by triple quadrupole and measured for its signal intensity (CPS). The mass spectrometry data was analyzed using MultiQuant software and the chromatographic peaks were integrated and corrected. The peak area (Area) of each chromatographic peak represents the relative abundance of the corresponding compound.

To ensure the accuracy of qualitative and quantitative analysis, the mass spectrum peak of each metabolite in different samples was corrected based on retention time and peak distribution information. Figure R4 illustrates the integral correction results from a randomly selected metabolite in the samples.

A

B

Figure R1: The total ion current diagram detected by mass spectrometry of QC samples. A, positive ion. B: negative ion. The X-axis shows the Retention time (Rt) from metabolite detection, and the Y-axis shows the ion flow intensity from ion detection (intensity unit: CPS, count per second).

A

B

Figure R2: The total ion current overlapped diagram detected by mass spectrometry of QC samples. A, positive ion. B: negative ion. The X-axis shows the Retention time (Rt) from metabolite detection, and the Y-axis shows the ion flow intensity from ion detection (intensity unit: CPS, count per second).

A

B

Figure R3: Multiple reaction monitoring (MRM) metabolite detection multi-peak diagrams of ion flows of QC samples detected by mass spectrometry. A, positive ion. B: negative ion. The characteristic ions of each compound were selected by triple quadrupole and measured for its signal intensity (CPS). The mass spectrometry data was analyzed using MultiQuant software and the chromatographic peaks were integrated and corrected. The peak area (Area) of each chromatographic peak represents the relative abundance of the corresponding compound. The X-axis shows the Retention time (Rt) from metabolite detection, and the Y-axis shows the ion flow intensity from ion detection (intensity unit: CPS, count per second).

A

B

Figure R4: Integral correction diagram for quantitative analysis of ion of randomly selected metabolites in the samples. A, positive ion. B: negative ion. The X-axis of each sub-plot is the retention time (min), and the Y-axis of each sub-plot is the ion current intensity (CPS) of a certain metabolite ion detection.

3. The identification of the presence of neohesperidosides and the key gene need to double check and more functional evidence. the main12RhaT is nonfunctional in loose-skin mandarins due to SNP or frame-shifting, and other genes with 12RhaT function were almost not expressed. Thus, the presence of the neohesperidosides should be barely detected, therefore the function of CZG-jg206 as 12RhaT should not be possible. Finally, if the authors want to argue on the issue, pls provide data from in vitro enzyme characterization or functional evidence, correlation analysis is not enough and may mislead the authors.

Response: Thanks for raising this point. CZG_jg206 is the only gene show 78.9% similarity and homology with *Cm1,2RhaT* in *C. chachiensis* genome assembly. To test our hypothesis, we conducted experiments to the validate function of the target protein of CZG_jg206. The coding sequence of CZG_jg206 was cloned into pMAL vector, and the recombinant expression vector 206-PMAL was constructed and transferred into protein-expressing strain *Rosetta* to induce the expression of the target protein. However, the CZG_jg206 recombinant protein could not catalyze hesperidin-7-O-glucoside and naringin-7-O-glucoside substrates into neohesperidin and naringin, respectively (Figure R5 and

Figure R6). We have corrected the corresponding content and remove the description related to the functions of CZG_jg206.

However, it is important to note that naringin and neohesperidin were detected in *C. chachiensis* using different methods in various studies. Naringin has been detected in peel, pith, endocarp, pulp and seeds of *C. chachiensis* fruit (Sun et al., 2010). Neohesperidin and naringin were identified in Guangchenpi (the dried mature peel of *C. chachiensis*) using UPLC-QTrap-MS/MS technology (Liang et al., 2022) and UPLC-Q-Exactive Orbitrap/MS (Sun et al., 2023), as well as by HPLC and HPLC combined with ion-trap and time-of-flight mass spectrometry (LC/MS-IT-TOF) (Zheng et al., 2020). Yu et al reported that neohesperidin was one of the 20 bound flavonoids of Guangchenpi identified by HPLC-Q-TOF-MS/MS (Yu et al., 2022). These results indicate the presence of naringin and neohesperidin in the fruit of *C. chachiensis*, but the mechanisms underlying the production of these neohesperidosides remain unknown.

Furthermore, CZG_jg16074 exhibits 99.86% similarity with *Crc1,6RhaT* which have been characterized as hesperidin synthase gene in *C. chachiensis*, and *Crc1,6RhaT* could transform flavanone-7-*O*-glucoside into hesperidin (Shang et al., 2022). These findings suggested that CZG_jg16074 might have the similar function. We updated the content in line 417 to 421 as follows.

"Interestingly, *Crc1,6RhaT* cloned from *C. chachiensis* recently which showed 99.86% sequence similarity with CZG_jg16074 has been identified as hesperidin synthase gene and *Crc1,6RhaT* could transform flavanone-7-*O*-glucoside into hesperidin⁴⁵, indicating that CZG_jg16074 might have a similar function."

References:

- [1] Liang, et al. Study on Flavonoids and Bioactivity Features of Pericarp of *Citrus reticulata* "Chachi" at Different Harvest Periods. *Plants (Basel)*. 2022;11(23):3390.
- [2] Sun, et al. Simultaneous determination of flavonoids in different parts of *Citrus reticulata* 'Chachi' fruit by high performance liquid chromatography-photodiode array detection. *Molecules*. 2010;15(8):5378-5388.
- [3] Sun, et al. Flavonoids contribute most to discriminating aged Guang Chenpi (*Citrus reticulata* 'Chachi') by spectrum-effect relationship analysis between LC-Q-Orbitrap/MS fingerprint and ameliorating spleen deficiency activity. *Food Sci Nutr*. 2023;11(11):7039-7060.
- [4] Yu, et al. Aged Pericarpium *Citri Reticulatae* 'Chachi' Attenuates Oxidative Damage Induced by tert-Butyl Hydroperoxide (t-BHP) in HepG2 Cells. *Foods*. 2022;11(3):273.
- [5] Zheng et al. Construction and Chemical Profile on "Activity Fingerprint" of *Citri Reticulatae* Pericarpium from Different Cultivars Based on HPLC-UV, LC/MS-IT-TOF, and Principal Component Analysis. *Evid Based Complement Alternat Med*. 2020;2020:4736152.
- [6] Shang, et al. *Crc1,6RhaT* is involved in the synthesis of hesperidin of the main bioactive substance in the *Citrus reticulata* 'Chachi' fruit, *Horticultural Plant Journal*, 2023;2468-0141.

Figure R5: Gel picture of CZG_jg206 recombinant protein. P: purified CZG_jg206 recombinant protein. U: Unpurified supernatant after IPTG induction. M: Protein marker.

Figure R6: HPLC analysis of enzyme activity reaction of CZG_jg206 recombinant protein. NC: negative control.

Reviewer #2 (Remarks to the Author):

The resubmission by Wen et al provided additional experimental data and re-organized context; by which substantial improvement was made for the readability and clarity of the manuscript. The authors have addressed the most of my concerns.

I feel it might be better that the NC editorial office staff could help to further work on the English writing of the manuscript and to make more concise description/presentation.

A few minor issues:

1. **Line 84 “(CCoAOMT), which needs ion catalyzes in the reaction” might change to “which needs ionic cofactors in the reaction”**

Response: Thanks for your comment. We have corrected it.

2. **please note that O-methyltransferase is to transfer methyl group to the hydroxyl moiety of substrate, it is not to add the “methoxyl” group. Such error occurs throughout the manuscript, for examples,
Line 213, “3'-OH methoxylation”
Line 304, “by an OMT adding the O-methoxy group into”
Line 545, “can directly add the O-methoxy group to the 3-OH site”**

Response: Thanks for your comment. We have corrected them.

3. **In addition, O should be italic in O-methylation or O-glycosylation, O-methyltransferase/-glucosyltransferase, and in the compound names.**

Response: Thanks for your comment. We have corrected them.

Reviewer #1 (Remarks to the Author):

The authors addressed my questions in a detailed and reasonable way. I thus recommend publication of this manuscript.

One minor issue is that the marker information are necessary to be included in the Figure R5.

Reviewer #2 (Remarks to the Author):

Although the authors have addressed most of my concerns, I echo reviewer 1's concerns and do not feel the authors have properly addressed those questions.

On flavonoid identification, the molecular ion fragment information is necessary to be provided as extended data---I think this was also my request in the first-round review. The authors presented total ion chromatography in this revision; however, they are not very useful.

The extremely low catalytic efficiency of CcOMT1 indeed is a concern on whether it really contributes to the formation of HPMF in planta.

Furthermore, within this revision, the evidence of functional validation of identified UGT vanished. The claimed identifications of UGT and regulators involved in PMF biosynthesis were purely based on data correlation but lack solid experimental validation, which makes part of conclusion unreliable.

Overall, the omics studies in this manuscript are sound but the evidence on functional identifications of OMT, UGT and TFs is weak.

Dear Reviewers,

We greatly appreciate your constructive comments and suggestions. We have responded to your specific queries in the following sections.

REVIEWERS' COMMENTS

Reviewer #1 (Remarks to the Author):

The authors addressed my questions in a detailed and reasonable way. I thus recommend publication of this manuscript.

One minor issue is that the marker information are necessary to be included in the Figure R5.

Response: Thank you for your comments. We have added the marker information to the figure and refined the Figure R5 below.

Figure R5. Gel image of the CZG_jg206 recombinant protein. **P:** purified CZG_jg206 recombinant protein. **U:** Unpurified supernatant after IPTG induction. **M:** protein marker.

Reviewer #2 (Remarks to the Author):

Although the authors have addressed most of my concerns, I echo reviewer 1's concerns and do not feel the authors have properly addressed those questions.

On flavonoid identification, the molecular ion fragment information is necessary to be provided as extended data---I think this was also my request in the first-round review. The authors presented total ion chromatography in this revision; however, they are not very useful.

Response: Thank you for your comments. For flavonoid identification, we obtained tandem mass spectra for 24 of 29 polymethoxylated flavonoids (see Extended Data Figure 5 to 16 below), and the remaining 5 polymethoxylated flavonoids do not have secondary ion mass spectra but their retention time aligns with the data in the database. We also updated the Supplementary Data 8 (previous Table S8) "The relative content of PMFs in fruits and peels of *C. chachiensis*." with compound retention time and molecular weight. Since the LC-MS/MS data were processed by a qualified commercial company (METWARE, Wuhan, China), they refused to provide all flavonoid compound retention time because their self-construction database was used for commercial purposes. However, we also uploaded all the LC-MS/MS data from this study to MetaboLights. Researchers can still use our data according to their own research interests.

The extremely low catalytic efficiency of CcOMT1 indeed is a concern on whether it really contributes to the formation of HPMF in planta.

Response: Thank you. We observed the activity of CcOMT1 *in vitro*, and the alterations of gene expression and accumulation of HPMF in transient overexpression and virus-induced gene silencing experiments. Considering the low catalytic efficiency of CcOMT1, we have revised the conclusion that CcOMT1 is a candidate enzyme involved in HPMF formation but not a key enzyme. We refined our manuscript accordingly and removed similar descriptions such as "key enzyme" and "important role".

Furthermore, within this revision, the evidence of functional validation of identified UGT vanished. The claimed identifications of UGT and regulators involved in PMF biosynthesis were purely based on data correlation but lack solid experimental validation, which makes part of conclusion unreliable.

Overall, the omics studies in this manuscript are sound but the evidence on

functional identifications of OMT, UGT and TFs is weak.

Response: To confirm the functional of the identified UGT, we have conducted experiments to validate the function of the target protein *CZG_jg206* and the details are provided in the last response letter. However, *CZG_jg206* did not present the expected function and we thus removed related descriptions. But based on correlation analysis results, some expected functions of the genes could be predicted. For example, *Crc1,6RhaT*, which was recently cloned from *Citrus reticulata* cv. Chachiensis and showed 99.86% sequence similarity with *CZG_jg16074*, has been identified as a hesperidin synthase gene and *Crc1,6RhaT* can transform flavanone-7-O-glucoside into hesperidin (Shang, et al., 2023). Although we could not completely rely on the correlation analysis results, they still provide insights for us to identify potential genes and validate the functions of these genes.

Therefore, we believe that the correlation analysis results of OMTs, UGTs and regulators (such as transcription factors) still provide valuable information for future studies.

Reference:

[1] Shang, N. et al. *Crc1,6RhaT* is involved in the synthesis of hesperidin of the main bioactive substance in the *Citrus reticulata* 'Chachi' fruit. *Hortic. Plant J.* (2023) doi:10.1016/j.hpj.2022.10.012.

The Extended Data Figure 5–16 are listed as bellows.

a

b

Extended Data Figure 5. Tandem mass spectra for a) 3,5,6,7,8,3',4'-Heptamethoxyflavone and b) Natsudaïdain-3-O-(3-hydroxy-3-methylglutarate)glucoside.

a

b

Extended Data Figure 6. Tandem mass spectra for a) 5,6,7,4'-Tetramethoxyflavanone and b) Gardenin B.

Extended Data Figure 7. Tandem mass spectra for a) 5,7-Dihydroxy-6,3',4',5'-tetramethoxyflavone and b) 3',4',5',5,7-Pentamethoxyflavone.

Extended Data Figure 8. Tandem mass spectra for a) 3'-Demethylnobiletin and b) 5-Hydroxy-6,7,3',4'-tetramethoxyflavanone.

a

b

Extended Data Figure 9. Tandem mass spectra for a) 3'-hydroxy-5,6,7,4'-tetramethoxyflavone and b) Tangeretin.

Extended Data Figure 10. Tandem mass spectra for a) Nobiletin and b) Chrysofenetin.

Extended Data Figure 11. Tandem mass spectra for a) 5,7,8,4'-Tetramethoxyflavone and b) Casticin.

Extended Data Figure 12. Tandem mass spectra for a) Sinensetin and b) 3',4',5,7-Tetramethoxyflavone.

a

b

Extended Data Figure 13. Tandem mass spectra for a) Natsudaidain and b) 5-Hydroxy-6,7,3',4'-tetramethoxyflavone.

Extended Data Figure 14. Tandem mass spectra for a) 7-Hydroxy-3,5,6,8-tetramethoxyflavone and b) 5,6,7,8,3',4'-Hexamethoxyflavanone.

Extended Data Figure 15. Tandem mass spectra for a) Natsudaidain-3-O-(5'-glucosyl-3-hydroxy-3-methylglutarate)glucoside and b) Isosinensetin.

Extended Data Figure 16. Tandem mass spectra for a) 5-Hydroxyauranetin and b) 5,4'-Dihydroxy-3,6,7,3'-tetramethoxyflavone-4'-O-glucoside.